# The Spatiotemporal Evolution and Influencing Factors of the Chinese Cities’ Ecological Welfare Performance

**DOI:** 10.3390/ijerph191912955

**Published:** 2022-10-10

**Authors:** Can Zhang, Jixia Li, Tengfei Liu, Mengzhi Xu, Huachun Wang, Xu Li

**Affiliations:** 1School of Government, Beijing Normal University, Beijing 100875, China; 2School of Business Administration, The Open University of China, Beijing 100039, China; 3China Life Reinsurance Company Ltd., Beijing 100039, China

**Keywords:** ecological welfare performance (EWP), regional differences, influencing factors, Theil index, spatial Durbin model

## Abstract

In the “full world” where natural capital is scarce, within the limits of the ecological environment, the improvement of welfare is a fundamental requirement for sustainable development. The ecological wellbeing performance (EWP) of 284 cities in China from 2007 to 2020 was measured by the superefficient SBM-DEA model, considering undesirable output, and analyzing the evolutionary trends of overall comprehensive technical efficiency, pure technical efficiency, and scale efficiency. The Theil index was used to explore the source and distribution of the Chinese cities’ EWP differences. Exploratory spatial data analysis (ESDA) and the spatial Durbin model (SDM) were applied to analyze the spatial distribution characteristics and driving factors of cities’ EWP. The results showed the following: (1) Regarding spatial and temporal distribution, the EWP of Chinese cities showed a fluctuating upward trend, in which pure technical efficiency > scale efficiency. (2) Considering regional differences, the differences in cities’ EWP were mainly intraregional rather than interregional. The contribution rates of distinct regions to the differences in EWP varied, i.e., western region > eastern region > central region > northeastern region. (3) In terms of spatial correlation, China’s EWP showed positive spatial correlation, i.e., high–high agglomeration and low–low agglomeration. (4) Concerning influencing factors, the level of financial development, the structure of secondary industries, the level of opening-up, and the degree of urbanization significantly improved EWP. Decentralization of fiscal revenue significantly inhibited improvement of EWP. Decentralization of fiscal expenditure and technological progress had no significant impact on the EWP. In the future, to improve cities’ EWP, China should focus on reducing differences in intraregional EWP, overcoming administrative regional limitations, encouraging regions with similar locations to formulate coordinated development plans, promoting economic growth, reducing levels of environmental pollution, and paying attention to the improvement of social welfare.

## 1. Introduction

Since China’s reform and opening up, rapid economic and social development has been accompanied by resource depletion, environmental pollution, and ecological degradation, which have seriously restricted the sustainable development of China’s economy and society [1]. Sustainable development is a comprehensive indicator to measure economic, environmental, and social inclusiveness, and is the key to national competitiveness [2,3]. Economic efficiency, ecological efficiency, or energy efficiency cannot fully reflect sustainable development [4]. However, ecological welfare represents the unity of economic efficiency, environmental benefits, and human welfare. It overcomes the limitations associated with the traditional measurement of welfare level by GDP growth, and can more comprehensively reflect the level of sustainable development [5], which is more in line with the principle of strong sustainable development [6]. The improvement of ecological welfare performance (EWP) is at the core of economic growth, ecological protection, and the improvement of people’s livelihoods. At present, China’s EWP level remains low. At the level of economic development, according to the open data provided by the International Monetary Fund (IMF), China’s per capita GDP in 2021 was USD 12,359, ranked 63rd in the world, still far from developed countries. In terms of environmental protection, pollutants generated by economic activities have become the bottleneck restricting China’s economic development. China’s carbon emissions remain high, while sulfur dioxide emissions and water pollution rates are not encouraging [7]. Regarding welfare promotion, a series of social problems including poverty and unemployment, along with concerns about education and health, have emerged in the process of urbanization, and have restricted China’s economic development [8]. These problems limit the sustainable development of China’s social economy [9], and reduce Chinese EWP [10].

The purposes of this paper are: (1) to establish an index system of EWP and measure Chinese cities’ EWP; (2) to explore the temporal and spatial evolution of Chinese cities’ EWP and the main sources of EWP differences; (3) to explore the spatial correlation and driving factors of Chinese cities’ EWP. The main contents of this paper are as follows: the next section summarizes the existing EWP research and identifies the research gaps. The third section introduces the construction of the index system, the methods, and describes the data sources in detail. The fourth explores the spatial and temporal evolution of Chinese EWP, the sources of differences, and the driving factors of Chinese cities’ EWP, by using exploratory spatial data analysis and spatial econometric models. The final part is of this paper includes the discussion and conclusion.

The significance of this paper lies in its correction of the measurement problems encountered in previous studies, achieved by rebuilding the input–output index system for EWP. At the same time, the research scale was refined to the city level to explore spatial and temporal evolutionary characteristics, regional differences, and the influencing factors affecting Chinese cities’ EWP, to enrich the relevant research, and provide a basis for the improvement of EWP in cities in China.

## 2. Literature Review

The implications of sustainable development are wide-ranging. EWP belongs to the category of sustainable development and was originally proposed to evaluate the sustainable development of cities [11]. Daly stated that human society has transitioned from an “empty world” rich in natural capital to a “full world” constrained by the ecological environment, and that natural capital has become a scarce factor restricting the development of human society [12]. Recognizing the natural limit of economic growth and the relationship between the ecosystem and the economic system, sustainable development economics includes focus on the improvement of welfare levels, and its prominence has risen rapidly. Ecological efficiency and economic performance are aspects of sustainable development. Although they are similar to EWP, these concepts remain the research paradigms of weak sustainable development. Focusing only on economic growth according to unit resource input cannot comprehensively measure sustainable development [13,14]. EWP is an upgraded version of ecological efficiency, and is a more relevant measure than ecological efficiency because it takes into account natural environment constraints and social welfare within the context of traditional economic efficiency guidance [15]. Based on the evaluation of ecological efficiency, EWP is a measure of integrated human development, connecting the three major systems of economy, society, and ecology [16]. It emphasizes the developmental concept of decoupling ecological resource consumption from social welfare, overcoming the limitations of traditional GDP in measuring the quality of human life [17]. EWP can reflect levels of local governance and also people’s happiness [18], and it can provide a new perspective and an analysis tool for strong and sustainable urban development [19].EWP is a people-oriented sustainable development concept, which helps to achieve cleaner production and improve the integration and coordination of economic systems, ecological systems, and the social system [20]. It reflects whether a country or region is moving closer towards or further away from sustainable development, and provides an analysis tool for studying national and global sustainable development [21].

At present, research on EWP is in the development stage, and mainly focuses on the measurement of EWP. With the developing maturity of the EWP concept, existing studies have most frequently used the ecological footprint proportion method [22], stochastic frontier analysis (SFA) [23], and DEA [16], among other methods for measuring EWP. When studies began, the ratio method was widely used to measure EWP. Zhu et al. used the ratio of human development index and ecological footprint to measure EWP. As a component of the HDI, economic development has improved technology and strengthened the ability to deal with natural disasters, as well as increasing overall levels of human education, all of which can help to mitigate the consequences of ecological deterioration. Therefore, measuring human welfare with HDI can help to increase levels of human welfare [24]. The ratio method is suitable for analyzing independent and discontinuous objects, especially single projects and technologies, while the indicator system is suitable for analyzing the coordinated development of multiple systems [25].Increasing numbers of scholars have used DEA to measure EWP. The advantages of DEA over the SFA model are that it does not need to set an exact production function, and it can solve multi-input and multi-output problems. Therefore, the DEA model has become the most commonly used method for measuring EWP.

In terms of input indicators, existing studies have mainly selected energy consumption, land consumption, and water resource consumption [26]. Many studies have also included environmental pollution as an input indicator, and most studies have not included human capital or material capital inputs in the index system [27,28]. In terms of output indicators, existing studies have usually used levels of economic development, education development, and medical and health development [27,29], and have not included environmental pollution. Hu et al. regarded environmental pollution as an undesirable output [16,30,31]. Using the network DEA method, Hu et al. [16] measured the EWP of 41 cities in the Yangtze River Delta in China from 2001 to 2017. Hou et al. measured the EWP level of 30 provinces (autonomous regions and municipalities directly under central governance) in China from 2006 to 2017 [6]. Liu et al. measured the EWP level of 13 urban agglomerations in China from 2010 to 2019, based on data from 171 prefecture-level cities [32]. Using the ratio method, Zhang et al. measured the EWP of 82 developed countries in 2012, and found that the EWP level of most countries was still low [33]. Sweidan et al. measured the EWP level of Gulf countries, also using the ratio method. Constrained by the problem of data acquisition from prefecture-level cities in China, the existing studies have mainly focused on a few cities [19,27], provinces, and regions [20], or have been transnational in scale [34], and fewer studies have been carried out at city and county levels. In terms of the factors affecting EWP, existing studies mainly explore the factors affecting EWP from the aspects of economy, society, and government support. For example, Wang explored the impact of residents’ income levels, local industrial structure, and technological progress on EWP [28]. Silva explored the impact of affluence and income inequality on EWP [35]. Xu et al. explored effects on EWP resulting from the five aspects of innovation, coordination, greenness, openness, and sharing [36]. Few studies have used spatial measurement methods to analyze the impact factors of EWP. We list some EWP research in Table 1.

Research on EWP has made great progress in recent years, developing from concept to measurement and then to mathematical modelling supported by data [40]. Certain problems have been encountered in the construction of the index system, as most studies regard only ecological resources as input indicators, ignoring non-resource input. Meanwhile, many studies have included environmental pollution among the input indicators, leading to EWP measurement problems. Among the research literature, most of the studies described have considered EWP at the national or provincial level, and few have addressed EWP at the prefecture level [41]. China is currently facing the challenge of coordinated development of economic transformation and welfare promotion. To promote strong and sustainable development in China, it is necessary to reconstruct the EWP indicator system, measure performance levels, determine the sources of differences, and explore the path to improving cities’ EWP.

The possible contributions of this paper include the construction of an index system; most previous studies in this field regarded environmental pollution as the input, while human capital and material capital input were not included. To measure EWP more accurately, human capital and material capital were added to the input indicators, and environmental pollution was included in the undesirable output. Most of the existing research studies have measured EWP at the provincial, regional, or national level, and few studies have refined it to the city scale. This paper discusses the spatial evolution trends and influencing factors of EWP at the city level in China, which is helpful for refining the existing research. In terms of research method, few studies have included spatial factors when exploring the impact of EWP. This paper uses spatial exploratory data analysis and spatial econometric analysis to provide policy recommendations for improving EWP.

## 3. Index Construction, Methods, and Data Sources

### 3.1. Index Construction

EWP refers to the efficiency of converting unit resource consumption into welfare, which can reflect sustainable development under the constraint of ecological resources. Establishing a scientific, comprehensive, and reasonable indicator system is the key to measuring EWP. According to the definition proposed by Rosa et al., EWP is the efficiency of a country’s use of economic, natural, and human capital resources to improve the level of human welfare [42]. In terms of selection of input indicators, natural capital in the ecosystem directly or indirectly contributes to human welfare [43]. Most of the existing studies have included natural resource input, for example, water resource consumption, land resource utilization, and power consumption [44]. In addition to these indicators, this paper also includes labor and asset input within the indicator system. The reason for this is that EWP describes the efficiency of transforming natural consumption into welfare, during which the ecosystem can provide direct ecological services which can only be transformed into material wealth through a series of links such as production, processing, consumption, and distribution to meet the needs of human welfare [16].Xiao et al. also stated that ecological input should cover not only resource indicators, but also introduce capital indicators to comprehensively reflect regional EWP [45].The current study selected energy consumption, water resource consumption, land resource consumption, labor input, and property input, which were measured by social electricity consumption, water consumption, built-up area, numbers of environmental protection personnel, investment in fixed assets of public utilities, and environmental protection expenditure. For the input index, this paper selected not only the input of ecological resources, but also included labor and property [16]; it should be noted that this approach is different from most existing studies. A city’s public utilities are important parts of its infrastructure, which play an important role in ensuring the normal operation of the city and improving the living environment. The main investments include water supply, gas, rail transit, drainage, and landscaping, which affect the scale, speed and health of the city’s development [46], therefore, these were also included in the index.

In terms of output, our model divides desirable and undesirable output. According to Hu et al. [16], welfare can be divided into three categories: economic welfare, social welfare, and environmental welfare [47,48]. In the undesirable output, this paper not only includes three industrial waste products in the evaluation system, but also includes carbon dioxide emissions. Carbon dioxide is known as a greenhouse gas, and increases in its emissions are the main reason for global temperature rises. At the same time, carbon dioxide emissions are also considered a cause of health risks. Betti et al. regarded carbon dioxide emissions as an objective environmental variable affecting national welfare [49], Yasin et al. also measured environmental well-being based on carbon dioxide emissions [50]. In the selection of output indicators, this paper selected welfare level as the desirable output and environmental pollution as the undesirable output. Among these, welfare output includes economic welfare, social welfare, and environmental welfare, which were measured by cities’ GDP, green spaces, average years of education in the population, numbers of doctors, and total road area. Environmental pollution was measured by three kinds of industrial waste discharge, and carbon dioxide emissions. Considering global efforts to achieve carbon neutrality, it is necessary to include carbon dioxide emissions in the indicator system. It should be noted that adding variables to the DEA model will result in higher weighting of spatial dimensions and efficiency scores [51].That is, when the number of DMUs is fixed, the more input–output indicators included in the model, the greater the efficiency value obtained, and the lower the identification of DEA analysis [52].Therefore, the variable selection of the DEA model should include as few factors as possible, and the results of existing research should be used for reference [53]. In this paper, the entropy method was applied to enable the use of the combined pollution index of three industrial waste products along with carbon dioxide emissions as the undesirable output. The EWP indicator system is shown in Table 2:

### 3.2. Research Methods

Research methods are based on the applicability of research questions. This study first explored the spatio-temporal evolutionary trends of EWP, so we used the super-SBM-DEA method to measure cities’ EWP. The DEA method can solve the problems of inconsistent input of various resources and output units of environmental pollution, and does not need to consider specific production functions, weights, and parameters. It overcomes the limitation associated with the upper limit of traditional efficiency. When the DMUs reach the efficiency boundary, the efficiency values of the DEA can be reordered. After calculation, it was found that there were large differences in EWP between different cities, so we used the Theil index to observe the source of the differences in EWP. With the rapid economic growth after China’s reform and opening up, regional differences have become a major challenge in China. The regional balance of EWP is of great significance to China’s sustainable development. The commonly used indicators for assessing regional differences or inequality can be divided into three categories: coefficient of variation, Lorentz curve index, and entropy or information theory index. The coefficient of variation is easy to calculate, but is sensitive to outliers. The Gini coefficient index is based on the Lorentz curve, but is easily affected by high values. One of the advantages of the Theil index is that it can be decomposed into additive terms to describe the inequality between and within elements in the system [54]. The strength of the Theil index is that overall regional differences can be decomposed into interregional and intraregional differences, which can better reflect the regional imbalance in the heterogeneous regional structure [55]. To further improve levels of EWP it is necessary to explore the relevant factors, and the impact of spatial effect cannot be ignored. Therefore, our study used spatial exploratory analysis to test the spatial correlation of ecological welfare performance. Exploratory spatial data analysis (ESDA) is a spatial data analysis method that can intuitively and clearly represent the spatial correlation and aggregation of geographical elements, and is widely used in spatial analysis [56]. Generally, global spatial autocorrelation can be applied to represent the spatial correlation of geographical phenomena across a region, and local spatial autocorrelation reflects the spatial concentration of certain elements. Spatial measurement methods were used in this study to explore the factors that affect EWP, and to provide empirical evidence for improving EWP. The specific model is shown below.

#### 3.2.1. Super-SBM-DEA Model

Data envelopment analysis (DEA) was first proposed by Charnes et al. in 1978. It is a nonparametric analysis method for measuring the relative efficiency between inputs and outputs of multiple factors [57]. The traditional CCR and BCC models did not consider the relaxation of input and output, resulting in inaccurate efficiency values. In view of this problem, in 2001 Tone proposed a non-radial and non-angle SBM-DEA analysis method based on measurement of variable relaxation. The advantage of this method is that the efficiency changes with the input and output relaxation levels [58]. However, the SBM model retains the problem that while the calculated efficiency value is effective for multiple simultaneous decision-making units, the effectiveness of the decision-making units cannot be compared. To solve this problem, in 2002 Tone proposed the super-SBM model, which can evaluate and rank the effective units of the SBM model [59].
(1)$$\min p=\frac{1+\frac{1}{m}\sum_{m=1}^{M}s_{m}^{x}/x_{jm}^{t}}{1-\frac{1}{l+h}(\sum_{l=1}^{L}s_{l}^{y}/x_{jl}^{t}+\sum_{h=1}^{H}s_{h}^{b}/b_{jh}^{t}}\;s.t.\left\{\begin{array}{c} x_{jm}^{t}\geq \overset{n}{\underset{j=1,j\neq 0}{\sum }}\lambda_{j}^{t}x_{jm}^{t}+s_{m}^{x}\\ y_{jl}^{t}\geq \overset{n}{\underset{j=1,j\neq k}{\sum }}\lambda_{j}^{t}x_{jl}^{t}-s_{l}^{y}\\ b_{jh}^{t}\geq \overset{n}{\underset{j=1,j\neq k}{\sum }}\lambda_{j}^{t}y_{jh}^{t}+s_{h}^{b}\\ \lambda_{j}^{t}\geq 0,s_{m}^{x}\geq 0,s_{j}^{y}\geq 0,j=1,\ldots ,n\;\; \end{array}\right.$$

The DEA method can solve the problems associated with inconsistent input of various resources and different output units of environmental pollution, and it does not need to consider specific production functions, weights, or parameters. Therefore, this study adopted the SBM-DEA model proposed by Tone to measure the EWP of Chinese cities. To reflect the actual situation more accurately, this paper introduces the undesirable output into the model for calculation. Suppose there are decision-making units in the production system, and each decision-making unit has three input–output variables, namely, input, desirable output, and undesirable output. The specific calculation for Formula (1) is the EWP, representing respectively the input, desirable output, and undesirable output values during the period. $S_{m}^{x}$, $S_{l}^{y}$, $S_{h}^{b}$ represent slack variable input, desirable output and undesirable output, respectively. λ is the weight vector of the decision unit.

#### 3.2.2. Theil Index

In 1967, Theil proposed the entropy concept of information theory to calculate income inequality, i.e., the Theil index, which was subsequently applied to measure regional differences in high-quality development and other fields. The Theil index decomposes overall difference into intragroup and intergroup differences, which can more intuitively reveal the sources and change trends of regional differences. At present, it is mainly utilized to study income inequality and the use of resources such as energy. It is an inequality coefficient to evaluate the differences between and within regions. The Theil index applies the concepts of information and entropy to explore inequality and difference. Compared with the Gini coefficient and the mean value of logarithmic dispersion, the Theil index can more accurately describe the differences between and within regions [60]. The Theil index decomposes the overall difference into intraregional and interregional differences. The specific formula is as follows:(2)$$T=\frac{1}{k}\sum_{q=1}^{k}\left(\frac{S_{q}}{\bar{S}}\times \ln\frac{S_{q}}{\bar{S}}\right),$$
(3)$$T_{p}=\frac{1}{k_{p}}\sum_{q=1}^{k_{p}}\left(\frac{S_{pq}}{{\bar{S}}_{p}}\times \ln\frac{S_{pq}}{\bar{S_{p}}}\right),$$
(4)$$T=T_{w}+T_{b}=\sum_{p=1}^{4}\left(\frac{k_{p}}{k}\times \frac{\bar{S_{p}}}{\bar{S}}\times T_{p}\right)+\sum_{p=1}^{4}\left(\frac{k_{p}}{k}\times \frac{\bar{S_{p}}}{\bar{S}}\times \ln\frac{\bar{S_{p}}}{\bar{S}}\right),$$

In Formula (2), T represents the total difference of national EWP, and its range is [0, 1]. The smaller the Theil index, the smaller is the overall difference in the development of EWP. q indicates the city, k indicates the number of cities, $S_{q}$ indicates the EWP of city q, and $\bar{S}$ indicates the average value of the national EWP level. In Formula (3), $T_{p}$ represents the total difference of region, $k_{p}$ represents the number of cities in Zone p, $S_{pq}$ represents the EWP of region p city, and $\bar{S_{p}}$ represents the average cities’ EWP in region p. In Formula (4), the overall difference in the development of cities’ EWP is further decomposed into the intraregional difference Theil index $T_{w}$ and the interregional difference Theil index $T_{b}$. In addition, define $T_{w}/T$ and $T_{b}/T$ as the respective contributions of intraregional and interregional differences to the overall variance, and $\left(S_{P}/S\right)\times \left(T_{P}/T\right)$ as the contribution of regions to overall differences within the region. $S_{p}$ is the sum of the EWP levels of the cities in region p, and S represents the sum of the EWP development levels of the cities.

#### 3.2.3. Spatial Autocorrelation

According to the first law of geography, there is a certain correlation between everything, and the closer the distance, the stronger is the correlation. This study aimed to verify the spatial correlation between the EWPs of various regions in China. The Moran index was used to measure the spatial distribution characteristics of various regions, with a value range of [−1, 1]. When the Moran index was greater than 0, it indicated that the EWP had a positive spatial correlation, and the larger the value, the stronger was the positive spatial correlation. When the Moran index was less than 0, it indicated that there was a negative spatial correlation with EWP. The smaller the absolute value, the weaker was the negative spatial correlation. When the Moran index was 0, the EWP presented a random distribution, and there was no spatial correlation. The Moran index can generally be divided into the global Moran index and the local Moran index. The global Moran index was utilized to reveal the overall degree of correlation of spatial attributes between regions. The local Moran index revealed the spatial correlation between different regions, as shown in Formulas (5) and (6):(5)$$I=\frac{\sum_{i=1}^{n}\sum_{j=1}^{n}w_{ij}\left(y_{i}-\bar{y}\right)\left(y_{j}-\bar{y}\right)}{\left(\sum_{i=1}^{n}\sum_{j=1}^{n}w_{ij}\right)\sum_{i=1}^{n}{\left(y_{i}-\bar{y}\right)}^{2}},$$
(6)$$I_{i}=\frac{\left(y_{i}-\bar{y}\right)}{\frac{1}{n}\sum_{i=1}^{n}{\left(y_{i}-\bar{y}\right)}^{2}}\sum_{j\neq i}^{n}w_{ij}\left(y_{j}-\bar{y}\right),$$

In Formulas (5) and (6), n is the total number of regions, $y_{i}$ is the EWP of region i, $\bar{y}$ is the average value of the national EWP, and $\omega_{ij}$ is the weight matrix.

#### 3.2.4. Spatial Durbin Model

Compared with the traditional regression method, the spatial measurement method considers the spatial correlation and spatial dependence of samples. The most frequently used models include the spatial error model (SEM), the spatial autocorrelation model (SAR), and the spatial Durbin model (SDM). The SDM considers the influence of the lag factor of the dependent variable on the explained variable, and the spatial spillover effect of different factors on the explained variable. The specific models are as follows:(7)$$EWP_{i,t}=\rho WEWP_{i,t}+WX_{i,t}\gamma +\mu_{i}+\eta_{t}+\varphi_{i,t}$$
where ρ represents the spatial lag term coefficient, W represents the spatial weight matrix,$\;\mu_{i}$ and$\;\eta_{t}$ represent the individual fixed effect and the time fixed effect, respectively. $X_{i}$ represents the explanatory variable. γ represents the spatial autoregressive coefficient of the explanatory variable, and $\varphi_{it}$ represents the random interference term. The spatial weight matrix is the key to spatial measurement. This approach is based on Lesage’s simplification principle of spatial weight [61]. Adjacency weight is widely used compared with other spatial weight matrices, however, the matrix assumes that whenever there is an association between two regions, the degree of association is the same (equal weight). This assumption often runs contrary to common sense. In related research on environmental pollution and economic development, it is generally accepted that regions at relatively close distance have higher degrees of correlation, and it is appropriate to select the weight matrix of spatial distance accordingly. Therefore, our study adopted the inverse distance matrix as the benchmark spatial weight matrix for empirical analysis, and then constructed the proximity matrix (W) and the economic distance matrix (E) to test the robustness of the spatial measurement results. The inverse distance matrix (D) is the reciprocal of the distance between cities. In consideration of the attenuation of the spatial effect with increasing distance ($l_{ij}$), an inverse distance matrix was established based on the reciprocal of the square. If i≠j, then the weight $d_{ij}=1/l_{ij}{}^{2}$, otherwise $d_{ij}$ = 0. The geographical adjacency matrix assumes that the adjacency matrix consists of only 0 and 1. If the cities are adjacent to each other, the weight $\omega_{ij}$ = 1, otherwise $\omega_{ij}$ = 0. In addition, economic development between regions is interrelated and mutually affected. Per capita GDP was used as the indicator of economic development, and the economic distance matrix (E) was constructed. If i≠j, the weight is $e_{ij}=1/\left|g_{i}-g_{j}\right|$, otherwise, the weight is $e_{ij}=0$.

### 3.3. Data Sources and Regional Division

Data sources relating to EWP measurement and its influencing factors are shown Table 3. In terms of data sources, certain points require explanation; the first of these concerns the data collection of smoke and dust emissions and carbon dioxide emissions. In 2020, the statistical reporting system for these emissions was significantly adjusted, and the statistical caliber and other aspects were changed compared with the statistical reporting system used in the 13th five-year plan. Smoke and dust were renamed particulate matter. Second, regarding carbon dioxide emissions data, some scholars have tried to use nighttime light data to retrieve the carbon emission footprint of municipal or lower administrative regions. However, due to defects of the light data, such as background noise and discontinuity, this method has not been widely accepted. Carbon dioxide emissions data were taken from the open anthropogenic carbon dioxide (ODIAC) fossil fuel emissions dataset of the Global Environmental Research Center (https://db.cger.nies.go.jp/dataset/ODIAC/ (accessed on 15 April 2022)). ODIAC first introduced the combination of nighttime lighting data and the emission–location profile of a single power plant for estimating the spatial range of carbon dioxide emissions from fossil fuels. The spatial resolution is 1-KM, and the unit is t/KM2. The product is generated by combining multi-source nighttime lighting data, a global point source database, and ship or aircraft fleet tracking. The data can represent global, regional, and city-level CO_2_ emissions and meet the requirements of large-scale and long-term series [62]. The open-source data inventory for anthropogenic CO_2_ (ODIAC) data was the monthly dataset of carbon emission spatial grid data derived by ODA and other countries. The annual statistical data of countries were used for spatial and temporal decomposition to obtain the global monthly spatial grid data [63]. By cutting, synthesizing, and extracting China’s carbon emissions grid data, measurements of the carbon dioxide emissions of China’s prefecture level cities from 2007 to 2019 were obtained, and the carbon dioxide emissions in 2020 were supplemented using the trend function. In terms of data processing, this paper synthesizes four environmental pollutants into an environmental pollution index. For missing data in individual years, the trend function and the interpolation method were employed to fill the gaps. To eliminate the influence of price factors, the data related to prices in the full text were treated as constant based on 1978 prices.

The formula for calculating the average length of schooling was:$$\mathrm{AYS}=\frac{6\times P_{primary\;school}+9\times P_{junior\;high\;school}+12\times P_{senior\;high\;school}+16\times P_{junior\;college\;and\;above\;}}{P_{primary\;school}+P_{junior\;high\;school}+P_{senior\;high\;school}+P_{P_{junior\;college\;and\;above\;}}}$$, where p is the population at each educational level.

To reflect the social and economic development of different regions, the State Council of China has divided China’s economic regions into four: the east, the middle, the west, and the northeast (Figure 1). The eastern region includes Beijing, Tianjin, Hebei, Shanghai, Jiangsu, Zhejiang, Fujian, Shandong, Guangdong, and Hainan. The central region includes Shanxi Province, Anhui Province, Jiangxi Province, Henan Province, Hubei Province, and Hunan Province. The western region includes the Inner Mongolia Autonomous Region, Guangxi Zhuang Autonomous Region, Chongqing City, Sichuan Province, Guizhou Province, Yunnan Province, Tibet Autonomous Region, Shaanxi Province, Gansu Province, Qinghai Province, Ningxia Hui Autonomous Region, and Xinjiang Autonomous Region. Northeast China includes Liaoning Province, Jilin Province, and Heilongjiang Province. In this study, 284 cities were divided into eastern cities, western cities, central cities, and northeastern cities according to their different locations.

Figure 2 shows the flow chart of the study method. First, the EWP index system was constructed, and then the EWP of 284 cities was measured using the superefficient SBM-DEA model. To describe the spatiotemporal distribution characteristics of cities’ EWP, the time evolution trends of the decomposed comprehensive technical efficiency, pure technical efficiency, and scale efficiency were analyzed, then ArcGIS technology was employed to display the distribution characteristics of 284 cities’ EWPs.Then, the total and regional differences of EWP were calculated using the Theil index. To identify the factors affecting EWP and the path to improving EWP, considering the possible spatial correlation of EWP, global spatial correlation was utilized to assess whether EWP values demonstrated spatial regression or OLS regression, then LM and LR were used to test whether EWP fitted the SEM, SAM and SDM models, and then Hausman testing was undertaken to assess whether it complied with a fixed effect model a or random effect model. After exploring the factors that affect EWP, this paper proposes suggestions to improve cities’ EWP.

## 4. Results

First, the super-SBM-DEA model in Max DEA software was used to measure the cities’ EWP. Then, the EWP of 284 cities in China was analyzed by spatial statistical analysis and exploratory spatial data analysis. Finally, the factors influencing the EWP of Chinese cities were analyzed.

### 4.1. Temporal Evolution of China’s EWP

“EWP” can be divided into “pure technical efficiency” and “scale technical efficiency”. “Comprehensive technical efficiency” is the product of scale efficiency and pure technical efficiency. Comprehensive technical efficiency is the ability to maximize output under a given input, or to minimize input under a given output. Pure technical efficiency is affected by technology, and scale efficiency is affected by the resource allocation decisions of managers [64]. To better describe the changing trends of cities’ EWP, five types of efficiency and effectiveness are defined: [0, 0.4] represents strong inefficiency, [0.4, 0.7] denotes inefficiency, [0.7, 1.0] is weak inefficiency, [1.0, 1.3] refers to efficiency, and [1.3, 1.8] indicates strong efficiency. Considering that the natural breakpoint method in ArcGIS cannot directly compare EWP across different years, this study used the natural breakpoint method and the equal division method in ArcGIS to grade the ecological welfare performance. When EWP exceeded 1.0, it was effective; less than 1.0, it was invalid. In the analysis of the effective range, this paper uses 1.3 as the dividing point; values exceeding 1.3 were considered strongly effective, while those at or below 1.3 were weakly effective. In the analysis of the invalid range, 0.4 and 0.7 are the dividing points, less than 0.4 was strongly invalid, 0.4–0.7 invalid, and 0.7–1.0 was weakly invalid. In terms of comprehensive technical efficiency, EWP showed a fluctuating upward trend across the country. According to the average value of Chinese cities’ EWP from 2007 to 2020 (Figure 3), comprehensive efficiency fluctuated around approximately 0.85, and was weakly ineffective. From 2008 to 2009, the comprehensive efficiency of the cities’ EWP showed an upward trend, perhaps due to the national emphasis on pollution reduction during the 2008 Beijing Olympic Games and the strengthening of water and air pollution controls. Environmental quality was improved compared with the previous levels. After four years of steady development, Chinese cities comprehensive efficiency of EWP dropped from 0.87 to 0.84 in 2013, because China’s environmental pollution was at its most serious in 2013. After a three-year stable period, the comprehensive efficiency of EWP gradually increased in 2017 and reached 0.9 in 2020. This is mainly because the high-quality development proposed in 2017 by the 19th People’s Congress of China has become the basic direction of China’s economic development. The concepts of innovation, coordination, greenness, openness, and sharing have been integrated into regional economic and social development, and various regions have made progress in innovation-driven general prosperity and sustainable development.

From the perspective of efficiency decomposition, the pure technical efficiency of Chinese cities’ EWP was higher than their scale efficiency, and the change trends of the two were different. From 2007 to 2020, the “pure technical efficiency” fluctuated around approximately 0.95, showing an almost effective state overall. Specifically, pure technical efficiency showed a downward trend from 2007 to 2008, a fluctuating upward trend after 2008, a downward trend after 2013, and a fluctuating upward trend after 2017. Pure technical efficiency plays a prominent role in improving Chinese cities’ EWP, followed by scale efficiency. The contribution of technological progress to the improvement of EWP is greater than the scale agglomeration effect. At present, the main obstacle to the improvement of EWP is an unreasonable input scale, resulting in overall EWP in a weakly effective state. In the future, it will be necessary to further utilize advanced management experience to improve the efficiency of resource allocation.

### 4.2. Spatial Distribution of Cities’s EWP in China

To explore the spatial distribution of China’s EWP, the levels of EWP in different regions of China, including decomposition, are presented in Figure 4. To present more intuitively the distribution patterns of cities’ EWP in China, cities’ EWP values were divided into five grades using the natural breakpoint method in ArcGIS. Four years, i.e., 2007, 2010, 2015, and 2020 are presented for visual analysis (Figure 5).

#### 4.2.1. Regional Distribution

From the perspective of comprehensive technical efficiency, the EWP of the four regions was found to be weak and ineffective, and the differences were significant (Figure 4). Over time, the EWP of each region has fluctuated and risen (except for the eastern region). Specifically, the comprehensive technical efficiency of the eastern and western regions fluctuated within the interval 0.85–0.90, and the comprehensive technical efficiency of the central and northeastern regions fluctuated between 0.80–0.85 and 0.75–0.90, respectively. The reason for the distribution pattern of “western region > eastern region > central region > northeastern region” is that the ecological advantages of the western region can reduce dependence on natural resources, bring economic benefits, and promote the improvement of subjective and objective social welfare. Although the eastern region has advanced technology and a high level of economic development, as the region where the industrialization process first started, pollution has for a long time been a serious issue, and environmental pollution has offset the welfare gains of economic and social development. In recent years, the central region has received industrial transfer from the eastern region, and rapid economic development has brought serious environmental pollution problems. The long-term economic development of northeastern China has been relatively slow. For historical reasons, many heavily industrialized cities are distributed in the region, and environmental pollution is a serious concern. 

From the perspective of efficiency decomposition, a significant difference was found between the east and the west in terms of pure technical efficiency and scale efficiency of EWP. The overall characteristics follow the pattern of pure technical efficiency > scale efficiency. Specifically, pure technical efficiency in the eastern region fluctuated within the interval 0.950–1.00, and the scale technical efficiency fluctuated between 0.850–0.950. The pure technical efficiency of the western region fluctuated between 0.95–1.0, and its scale technical efficiency fluctuated between 0.90–0.95. This shows that in the western and eastern regions, technology promotion contributes a high proportion of EWP, and there remains room for further improvements in scale efficiency. It also suggests that the regions have not taken full advantage of scale and agglomeration, and that the mechanism of interregional cooperation is imperfect. In the future, regional cooperation can promote and improve the rational allocation of resources. No significant differences in pure technical efficiency and scale efficiency were observed between the central and western regions, and the values of pure technical efficiency and scale efficiency scores were alternately higher. The scale efficiency of the central region was higher than the pure technical efficiency in 2007–2013 and 2016–2017, and the pure technical efficiency was higher than the scale efficiency in other years. In 2009 and 2011–2012, the scale efficiency of northeast China was higher than the pure technical efficiency there, and in other years the pure technical efficiency was higher than the scale efficiency. Specifically, the scale technical efficiency of the central region fluctuated within the interval 0.90–0.95, the pure technical efficiency basically fluctuated within the interval 0.90–0.95, the scale technical efficiency of the northeast region fluctuated within the interval 0.85–0.95, and the pure technical efficiency fluctuated within the interval 0.85–1.0. The scale technical efficiencies of the four regions were less than 1, indicating that there were redundant inputs or insufficient outputs of ecological welfare resources.

#### 4.2.2. Cities’ Distribution

ArcGIS10.8 was utilized to describe the evolution characteristics of the Chinese cities’ EWP. The EWPs in 2007, 2010, 2015, and 2020 were selected as samples, and the natural break point method was applied to divide them into five types, i.e., strongly ineffective EWP, ineffective EWP, weakly ineffective EWP, effective EWP, and strongly effective EWP.

Figure 5 shows the spatial distribution of EWP over each of the four years. In the selected time range, within the general evolution trend of cities’ EWP, the change trends of different types of EWP are different. In general, the number of cities with ineffective EWP decreased, while the numbers of cities with weakly ineffective and effective EWP increased. Specifically, the number of cities where the EWP showed strong ecological ineffectiveness increased from one to three, namely Chongqing, Shanghai, and Tianjin. It can be seen that the economically developed municipalities directly under central government control exhibited serious resource input redundancies or output shortages. The line graph of the number of cities with ineffective EWP showed an inverted N-shape, from regional agglomeration to point distribution, decreasing from 89 in 2007 to 56 in 2010, then increasing to 82 in 2015, and then decreasing to 36. These cities were mainly concentrated in Beijing Tianjin Hebei, central and southern Liaoning, Harbin Great Wall, the West Bank of the Strait, the Great Bay area of Guangdong, Hong Kong and Macao, and the city agglomeration in the central reaches of the Yangtze River. The number of cities with weak and ineffective EWP showed an N-type change, rising from 68 in 2007 to 90 in 2010, and from 75 in 2015 to 88 in 2020, mainly concentrated in the city agglomerations in the central reaches of the Yangtze River and the agglomeration in the central plains. The number of cities with effective EWP also saw an N-type change, rising from 123 in 2007 to 136, and then from 126 in 2015 to 157. These were mainly concentrated in the Harbin Great Wall agglomeration, Hubao Eyu agglomeration, Guanzhong Plain agglomeration and Bei bu Gulf agglomeration. Cities with strongly effective EWP were scarce, with only one (Dongguan) in 2007 and 2010. There were no cities with strongly effective EWP in 2015 or 2020.

### 4.3. Regional Differences in China’s EWP

To further explore the regional differences and main sources of Chinese cities’ EWP, we used the Theil index and its decomposition model to measure regional and interregional EWP differences, and calculated the contribution rates of the four regions to the EWP differences for specific results (Figure 6 and Figure 7). Considering the overall value differences of cities’ EWP in China (Figure 6), the Theil index of the cities’ EWP in 2007–2020 was generally small, and presented a fluctuating downward trend, showing that the difference in overall EWP between cities has been narrowing over time. Specifically, it decreased from 0.034 in 2007 to 0.018 in 2020, with an average annual decrease of 0.1%. This is due to the large differences in economic, social, and environmental resource endowments between different regions. Reduction of the differences in EWP between regions represents a long-term project. From the decomposition results of the cities’ EWP differences, it was revealed that the main difference in cities’ EWP in China is a result of intraregional differences rather than interregional differences. During the period from 2007 to 2020, the intraregional differences in Chinese cities were always far greater than the interregional differences. The EWP difference within the regions fluctuated in the range of 0.018–0.033, showing a downward trend, indicating that the gap in EWP within China was narrowing. The interregional EWP difference remained low, and its value fluctuated within the range of 0.001–0.002. This also showed a fluctuating downward trend, indicating that the EWP difference between regions in China is small and the gap continues to narrow. From the decomposition results of EWP differences, it can be seen that the key to improving China’s EWP lies in narrowing the differences between cities within the region.

To further explore the sources of the differences in Chinese cities’ EWP, we analyzed the contributions of the four regions to the differences in EWP (Figure 7). It was found that the western region and the eastern region alternately made the highest contribution to the overall difference, followed by the central region, and the northeast region made the smallest contribution to the difference in EWP. The contribution rate of the eastern region fluctuated between 25% and 35%, and that of the western region fluctuated between 30% and 36%. The contribution rate of the central region fluctuated between 20% and 27%, and the contribution rate of the northeast region fluctuated between 9% and 15%, representing the smallest contribution to the regional difference. Specifically, between 2007–2011 and 2017–2018, the contribution rate of the western region to the difference was higher than that of the eastern region, ranking first among the four regions. In 2012–2016 and 2019–2020, the contribution rate of the eastern region to the difference was higher than that of the western region.

According to the decomposition results of the Theil index and the contribution rates in different regions, although there was little difference in EWP between regions, there were large differences in EWP within regions. From the contribution rates of different regions to the difference in EWP, it was seen that the eastern and western regions contributed nearly 70% to the difference in EWP. Therefore, China should in future focus on reducing the difference in EWP between the western and eastern regions, and promoting the coordinated development of cities. In particular, cities in the western and eastern regions should promote the coordinated improvement of EWP.

### 4.4. Spatial Autocorrelation of Cities’s EWP in China

#### 4.4.1. Global Spatial Autocorrelation Analysis

Table 4 shows the spatial correlation analysis using the global Moran’s I index and the inverse distance matrix, with test results obtained using Stata 15.1 software. The global Moran index value of Chinese cities’ EWP was significantly positive at the levels of 1% and 5% (except in 2018 and 2019), indicating that the spatial distribution of Chinese cities’ EWP was not random, but had significant spatial correlations and spatial clustering characteristics. This means that the EWP of a city was only affected by its own environment, but also by the surrounding cities. Specifically, cities with high EWP were surrounded by other cities with high EWP, and cities with low EWP were surrounded by other cities with low EWP. According to the overall time evolution trend, the Moran index of Chinese cities’ EWP showed a fluctuating downward trend, from 0.042 in 2007 to 0.008 in 2020, indicating that the spatial correlation of China’s cities’ EWP has been weakening.

#### 4.4.2. Local Spatial Autocorrelation Analysis

Although global spatial autocorrelation analysis can reflect the spatial clustering characteristics of Chinese cities’ EWP, it cannot reflect the internal relations between cities or the specific spatial correlation characteristics. Therefore, it was necessary to further test the local spatial autocorrelation characteristics of Chinese cities’ EWP.

To judge whether the cities’ local association was statistically significant from a quantitative perspective, we assessed the results intuitively from the significance level with the help of the spatial local autocorrelation index “LISA” (Local Indicators of Spatial Association). Geoda software was used to calculate and draw the LISA agglomeration map of Chinese cities’ EWP in 2007, 2010, 2015, and 2020. As shown in Figure 8, indicating significant changes in the number of cities in the LISA agglomeration map, the number of High–High (H-H) agglomeration cities presented a V-shaped change, the number of High–Low (H-L) agglomeration cities presented an inverted V-shaped change, and the numbers of Low–High (L-H) and Low–Low (L-L) agglomeration cities presented an inverted N-shaped change. Regarding the agglomeration areas, the H-H agglomerations were in the Beibu Gulf and Guan Zhong Plain. The L-L agglomerations were in central and southern Liaoning, Beijing, Tianjin, and Hebei. Over time, the L-L agglomeration areas gradually decreased and gradually shifted to the Yangtze River Delta cities, from regional agglomeration to point dispersion. The H-L agglomerations were mainly in the vicinity of Hubao, Eyu, central and southern Liaoning, and Harbin Great Wall, moving to the Yangtze River Delta over time. The L-H agglomerations were in Chengdu Chongqing and Guangdong Hong Kong Macao. 

### 4.5. Influencing Factors of Cities’s EWP in China

Using the global Moran index and the local Moran index in the spatial exploration analysis, a spatial correlation of Chinese cities’ EWP was found. In order to further reveal the influencing factors of Chinese cities’ EWP, Stata software was employed for regression using the spatial econometric model.

#### 4.5.1. Variable Selection and Interpretation

Taking the EWP of 284 cities in China from 2007 to 2020 as the explained variable y, this study analyzed the effect of each explanatory variable on the EWP of Chinese cities, thereby revealing the main factors affecting EWP levels. This paper selects the following explanatory variables in agreement with existing studies (Table 5):

Financial development level. The essence of finance is to integrate idle funds in society, so that these funds can be transferred from savings to long-term investment, which can ease the financial constraints of enterprises and improve the sustainability of enterprise innovation [65]. The improvement of EWP requires the coordinated improvement of economic growth and environmental protection. Financial development can provide financial support for environmental protection and economic growth, promote the rapid development of environmental protection and new technology, and thus have an impact on EWP. The balance of deposits and loans of financial institutions and the proportion of GDP (%) were utilized to measure the level of financial development.

Industrial structure. Different industrial structures correspond to different pollution structures and economic growth models, and the industrialization stage with high secondary production is often the fastest. Generally, the technological progress of secondary industry, especially manufacturing, is faster than that of the service industry, so it can more easily obtain a higher economic growth rate [66]. However, at the same time, a higher proportion of secondary industry leads to aggravation of environmental pollution. Therefore, the industrial structure affects EWP by affecting economic development and environmental pollution. The industrial structure was measured by the proportion of total output value of the secondary industry to GDP (%).

Fiscal revenue and expenditure decentralization. Fiscal decentralization essentially measures levels of financial resource distribution and financial incentives among governments at different levels. Fiscal decentralization may lead to local governments’ competitive destruction of the environment and have a negative impact on environmental protection. Local governments with more financial rights may be more inclined to promote economic growth in the context of competition. High level fiscal decentralization may aggravate environmental pollution and thus have an impact on EWP. The ratio of local level income to (central government income + local government income) and the ratio of local level expenditure to (central government expenditure + local government expenditure) were utilized to measure fiscal revenue and expenditure decentralization [67]. 

Innovation level. With traditional resource agglomeration and industrial production promoting cities’ development, the driving force of growth has gradually changed from factor agglomeration to technology promotion. Innovation has become the main driving force of cities’ development, and it is also the key to restraining the downward trend of China’s economic growth. Innovation promotes the growth of cities through the use of scientific and technological resources and innovation [68], with an impact on EWP. The innovation level was measured by the number of patents.

Level of opening up. Through intensive investment in innovation, foreign-funded enterprises have promoted China’s growth momentum from factor-driven to innovation-driven [69]. Furthermore, foreign investment has also created employment opportunities and expanded the scale of employment, which has led to the rise of national income. Moreover, the mature management systems and enterprise culture of foreign investors have guaranteed the welfare and income level of employees, which has an impact on EWP. However, there have been some negative effects of opening up, such as the crowding-out effect and structural imbalance effect. The level of opening up is measured by the proportion of foreign direct investment in GDP (%).

Urbanization level. Urbanization brings an agglomeration effect, promotes economic efficiency, and optimizes resource allocation. With the advancement of urbanization, the employment and income levels of residents, public facilities in cities and towns, and the quality of public services will be continuously improved [70]. The city–rural binary system has been broken, helping to reduce the income gap between cities and rural areas, while the improvement of people’s living conditions has improved welfare levels, bringing about improvements in EWP. The urbanization level is measured by the ratio (%) of the cities’ population to the total population.

#### 4.5.2. Analysis of Empirical Results

EWP can represent the sustainable development of a country or region, but it cannot guide the construction of sustainable development. It is necessary to further explore the influencing factors of EWP, to provide policy enlightenment for improving EWP.

First, it can be seen from the LM test results that both the spatial error and the spatial lag tests indicated a rejection of the original hypothesis (Table 6), suggesting that either the SLM model or the SEM model would be suitable, so the SDM model combining the two was selected. Then, according to LR test, it was found that according to the test results of the SDM and spatial autoregressive model, the original hypothesis was rejected; the SDM cannot be degenerated into a spatial error model or a spatial autocorrelation model, and the SDM was selected. Finally, according to the Hausman test, the conclusion was that the original hypothesis should be rejected and the fixed effect was preferable, so we selected the optimal SDM for spatial econometric analysis.

With the help of Stata 15.1, spatial econometric regression was conducted on the influencing factors of the EWP of 284 cities in China. Spatial metrology models include the spatial error model (SEM), spatial autocorrelation model (SAR), and spatial Durbin model (SDM). Through the above series of tests, this paper takes SDM model as the benchmark model to consider the impact of various variables on EWP. Three types of matrices were used to explore the factors affecting cities’ EWP (Table 7). The benchmark model used the spatial weight matrix constructed by the inverse distance matrix. To increase the robustness of the regression results, we further constructeed the proximity matrix and the economic distance matrix to estimate the benchmark model. Since the regression results were basically robust, here we focus only on the regression results under the inverse distance spatial weight matrix.

(1)The level of financial development had a significant effect on the cities’ EWP. The regression coefficient of the level of financial development was 0.026, significant at the 1% confidence level. The spatial lag coefficient was 0.186, significant at the 1% confidence level, indicating that the financial development level of neighboring cities can also drive the improvement of local EWP, and there was a spatial spillover effect. A high level of financial development means that enterprises can achieve good development by building a market-risk-sharing mechanism, thus driving economic growth and welfare levels and promoting cities’ EWP.(2)The regression coefficient of industrial structure to cities’ EWP was 0.001, significant at the 10% confidence level. The spatial lag coefficient was −0.008, but this was not significant. This shows that the secondary industry structure can effectively improve EWP. Although secondary industry brings environmental pollution, it has high production efficiency and rapid technological progress, which can quickly drive economic growth. Therefore, the proportion of secondary industry can promote EWP. The proportion of the secondary industry in neighboring cities had a negative impact on local EWP, but this was not significant.(3)The regression coefficient of fiscal revenue decentralization on cities’ EWP was −0.211, significant at the 1% confidence level. The regression coefficient of the spatial lag term was 1.363, significant at the 10% confidence level. This indicates that fiscal decentralization may lead to competition among local governments, cause environmental pollution, and then inhibit the improvement of EWP in local cities. However, fiscal revenue decentralization in neighboring cities does not affect local EWP. The regression coefficient of fiscal expenditure decentralization on cities’ EWP was −0.161, and the result was not significant. The spatial lag term was −0.446, and the result was not significant. This shows that the impact of local fiscal expenditure decentralization on local EWP, and on local EWP in neighboring cities, can be ignored.(4)The regression coefficient of innovation level on cities’ EWP was 0.002, and the result was not significant. The spatial lag regression coefficient was −0.067, and the result was not significant. This indicates that technological progress did not improve local EWP or that of neighboring cities. This shows that the current level of innovation has not played a full role in promoting the economy and reducing environmental pollution. In the future, to promote cities’ EWP it will be necessary to increase the research and development of green innovative technologies.(5)The regression coefficient of opening up on EWP was 0.292, which is significant at the 10% level, indicating that the hypothesis of “pollution paradise” is untenable. The spatial lag coefficient was 2.150, but was not significant. This shows that opening up can promote the local EWP of cities. Opening up can promote the EWP by promoting employment and improving welfare levels and economic development, but the opening up of neighboring cities did not significantly improve the local EWP.(6)The regression coefficient of urbanization on EWP was 0.002, significant at the 1% confidence level. The spatial lag coefficient was −0.006, but was not significant. This shows that urbanization can narrow the gap between cities and rural areas, drive economic growth, improve the level of welfare. and promote the improvement of local EWP. The spatial spillover effect was not significant, i.e., the urbanization of neighboring cities did not significantly inhibit the improvement of local EWP.

## 5. Discussion and Conclusions 

### 5.1. Discussion

EWP plays an important role in promoting economic, social, and environmental coordination. Within its framework design, this paper has further enriched the index system of EWP: (1) in terms of input indicators, in addition to incorporating existing natural resources [17,26,71], human capital and asset investment have been included; (2) in terms of output indicators, referring to existing research results [9,10,11], desirable output was divided into economic welfare, social welfare, and environmental welfare. At the same time, CO_2_ emissions have been included in the undesirable output as new pollutants. Meanwhile, the research scale has been refined to the city level, and the spatiotemporal evolution of cities’ EWP in China has been revealed in more depth and detail. 

The first research conclusion of this paper is that, on the level of time evolution, EWP is fluctuating and rising [6,26]. In terms of efficiency comparison, pure technical efficiency was found to be of greater significance than scale efficiency, which is consistent with the findings of Wang et al. [72].Secondly, in terms of spatial distribution, the results have similarities with those of existing provincial-level studies: the eastern region > the central region > the northeastern region [73]. Unlike the previous studies in which the western region ranked lower [26], the results of this study suggest that the EWP of the western region is higher than that of the eastern region. The reason EWP shows the trend of “the western region > the eastern region > the central region > the northeastern region” is that most of the western areas are ecologically fragile with better ecological environments. In recent years, the economy has developed rapidly and EWP has continuously improved. Although the eastern region is economically developed, it started industrialization earlier and the environmental pollution there is more serious. Therefore, the EWP of the western region is higher than that of the eastern region. As an important energy and industrial base in China, the central region has a long history of development, and its economy, science, education, and medical health levels are more developed than those in the west and the northeast. However, the central region continues to receive transfer of high pollution and high energy consumption industries from the east, making its advantages over the western region disappear, resulting in the situation that the EWP of the central region is lower than that of the east and the west [17,29].In recent years, the brain drain in northeast China has been serious, and internal resources are insufficient. As an important area of China’s old industrial base, its levels of environmental pollution are serious, making its EWP the lowest among the four regions. 

Third, at the level of regional differences, similar to the research of Wang et al. [74], it was observed that the overall difference in China’s EWP is narrowing [71]. However, intraregional differences are still considerable. These intraregional differences are the key to realigning the unbalanced development of regional EWP. In the future, focus should be concentrated on narrowing the range of cities’ EWP within the regions. The contributions of the east, west and, central regions to the differences in EWP have been mainly due to large economic differences. Overall regional differences have gradually decreased, mainly due to the gradual reduction of regional differences. Consistent with the research of Xu et al., the differences of EWP in northeast China are the smallest [75].Fourth, on the level of spatial correlation, these results were similar to the findings in the study by Hou [6], and this paper argues that there is a positive spatial correlation, with high–high clustering and low–low clustering, and the spatial clustering effect has shown a downward trend of fluctuation. Different from previous studies, high–high agglomeration of EWP was found to be concentrated in the Beibu Gulf and Guanzhong Plain agglomerations, which are not in economically developed areas. Low–low agglomeration was mainly in the eastern Beijing Tianjin Hebei agglomeration and the Yangtze River Delta agglomeration, which are economically developed regions. This spatial agglomeration feature does not conform to the basic pattern of regional economic development. Fifth, in terms of influencing factors, financial development level, secondary industry structure, opening up, and urbanization have significantly improved China’s EWP [74,76]. In the future, China should use the spillover effect of different influencing factors to promote the improvement of EWP. The limitation of this study was that in the construction of the index system this paper adopted the objective EWP evaluation method, ignoring subjective feeling and cognitive evaluation of individual perception. In the future, subjective and objective evaluation can be integrated into an index system to comprehensively evaluate cities’ EWP. This paper only applied spatial econometric analysis on the influencing factors of EWP, and did not further explore the path that affects EWP. In the future, we can select a certain aspect at the governmental or social level to further explore the specific path that affects EWP. We can also use the causal inference method to explore the net impact effect of a given policy on EWP, to furtehr enrich the research on EWP.

### 5.2. Conclusions

The super-SBM-DEA model considering undesired output was applied to measure the EWP of 284 cities in China. Spatiotemporal evolution characteristics and the mechanisms of cities’ EWP were analyzed by combining the Theil index, the exploratory spatial data analysis method, and the spatial econometric model. The following conclusions are drawn: 

Firstly, in terms of the time evolution of cities’ EWP in China, the cities’ EWP showed a fluctuating upward trend. The average EWP was in a weakly ineffective state, and the pure technical efficiency was higher than the scale efficiency, indicating that China should pay attention to improving EWP by improving management and optimizing resource allocation in the future. In terms of spatial change, differences of EWP and decomposition efficiency were observed in different regions, and it was found that in the terms of levels of comprehensive technical efficiency, the western region > the eastern region > the central region > the northeastern region. In the level of pure technical efficiency, the eastern region > the western region > northeastern region > the central region. In terms of scale and technical efficiency, the central region > the western region > the eastern region > the northeastern region. 

Secondly, the Theil index was employed to analyze the sources of differences in EWP. From the perspective of regional differences, differences in EWP were caused intraregionally, and the interregional contribution was small. The western region contributed most to the regional differences, followed by the eastern and central regions, and the northeastern region contributes the least. In future, we should focus on narrowing the EWP gaps in the western, eastern and central regions. Third, EWP showed a positive spatial correlation, with high–high agglomeration and low–low agglomeration characteristics. In terms of the factors affecting the cities’ EWP, under the weight of the spatial distance matrix, the use of the spatial Durbin model test found that financial development, the structure of secondary industries, levels of opening up, and urbanization rate all promoted EWP, while fiscal revenue decentralization inhibited the improvement of EWP. The impacts of technological progress and fiscal expenditure decentralization on EWP were not significant. 

### 5.3. Implications 

At present, China’s EWP is in a weakly ineffective state, indicating that there is room for improvement. In the future, it will be necessary to further improve EWP to an effective level. We need to transform the GDP-oriented development model, promote the conversion of new and old driving forces, and improve resource utilization by implementing the innovation-driven development strategy. Furthermore, we should pay attention to the improvement of social welfare, promote the equalization of public services, and improve people’s living standards. By improving management and technology, the scale technical efficiency, pure technical efficiency, and EWP will be improved.

For narrowing the regional differences in EWP, all regions should consider the development stage of the region and coordinate the improvement of EWP according to local conditions. The spatial correlation of EWP should be an important reference for the government to formulate development policies. Cities should actively cooperate with one another to overcome regional administrative barriers. While improving their own EWP, central cities will drive the development of other cities with low EWP. Cities with low EWP should consciously aim to benefit from the spillover effects of other cities, to improve their own EWP. The eastern region had the highest pure technical efficiency, with high-quality human capital and scientific research institutes that absorb international technology spillovers. The eastern region should focus on promoting social equity and coordinated social development. The central region should pay attention to industrial upgrading and transformation, introduce advanced technology, and make up for the shortcomings of people’s livelihoods. The western region should further expand the level of opening up on the basis of maintaining ecological advantages, and actively integrate into the Belt and Road initiative. Northeast China, which has the lowest EWP, should further improve the opening-up coordination mechanism and create a characteristic industrial chain to improve technology and management levels. 

Financial development levels, secondary industrial structure, opening up, and urbanization have significantly improved China’s EWP [74,76]. In the future, China should use the spillover effect of different influencing factors to promote the improvement of EWP.

Attention should be paid to the scale effect of finance and the promotion of technology spillovers. As financial development plays an obvious role in promoting EWP, all regions should speed up the reform of financial systems, constantly stimulate the innovation of financial products, leverage social capital into green production, and play a financial role in promoting EWP. The concept of green development should be applied throughout the whole process of industrial restructuring, to achieve sustainable development of the ecological environment and high-quality coordination of the economy. It is necessary to promote environmental protection and technological upgrading in secondary industry, and to increase productivity. China should vigorously develop new energy, new materials, high-end equipment manufacturing and other emerging industries, promote the green transformation of the manufacturing industry, and create a green and ecological industrial manufacturing cluster. Digital upgrading should be promoted to improve industrial structures, and the low-carbon transformation of industry should be promoted using digital technology. China should expand the level of opening up to the outside world, and adhere to the pattern of opening up from east to west and by land and sea. According to cities’ economic development levels, social and public needs, and regional factors, it should be possible to select the industrial attributes, investment scale, and equipment from foreign investment, implement industrial policy support and tax incentives for cleaner production enterprises, encourage the promotion of research and development technology, and give play to the scale and spillover effects of foreign investment. In the process of urbanization, local governments should adhere to the concept of green development, give consideration to the coordination of land expansion, infrastructure, and greening construction, and pay attention to the coupling and coordination of economic benefits and ecological efficiency. The new urbanization process should be accelerated at a county level, forming a new urbanization pattern with regional linkage, reasonable structure, which is intensive and efficient, green and low-carbon, and promotes the improvement of EWP.

## Figures and Tables

**Figure 1 ijerph-19-12955-f001:**
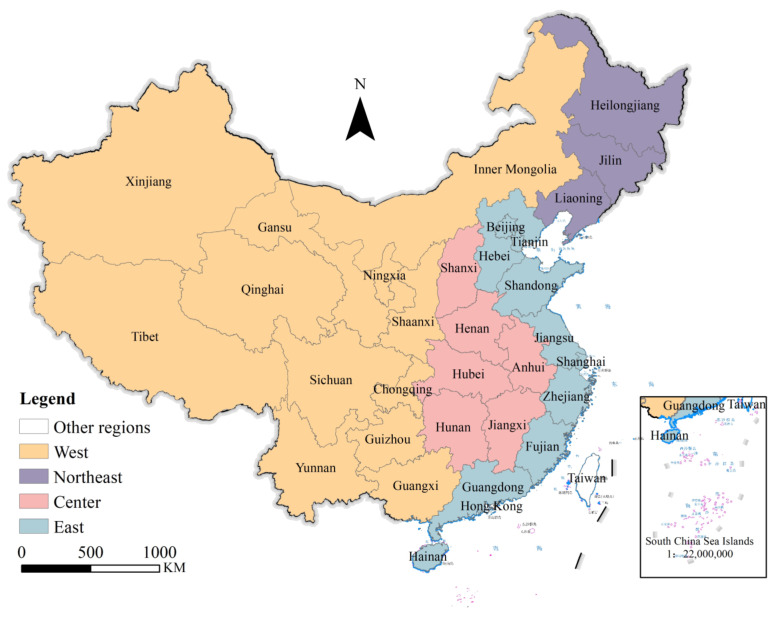
Division of four regions in China. Note: the map is drawn according to the Standard Map Service Website of the Ministry of natural resources (map review No. GS (2020) 4630).

**Figure 2 ijerph-19-12955-f002:**
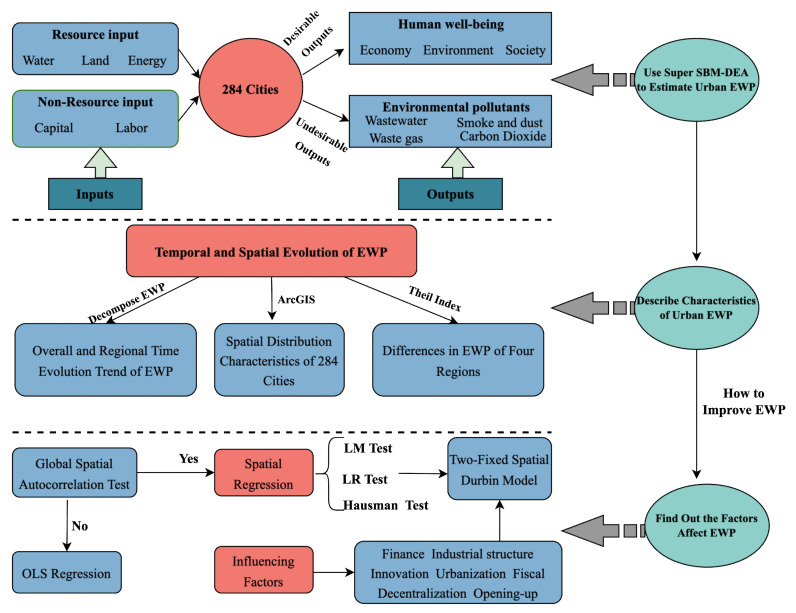
Research flow chart.

**Figure 3 ijerph-19-12955-f003:**
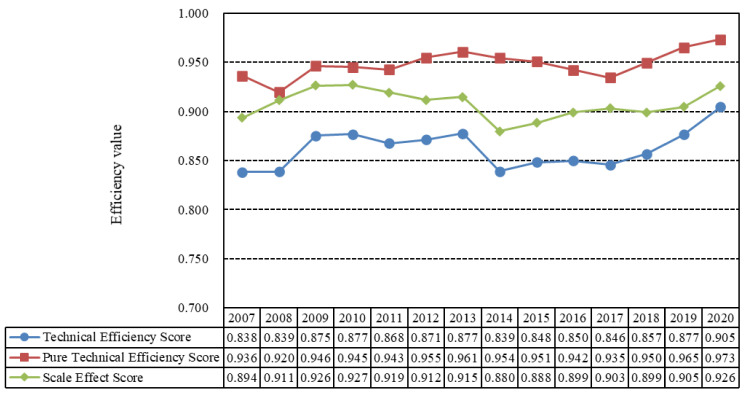
Temporal evolution of the cities’ EWP from 2007 to 2020, and its decomposition.

**Figure 4 ijerph-19-12955-f004:**
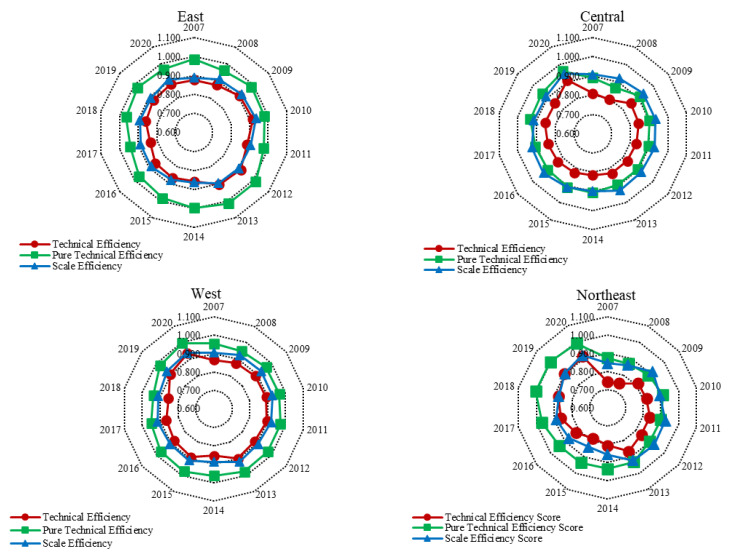
EWP of four regions in China from 2007 to 2020.

**Figure 5 ijerph-19-12955-f005:**
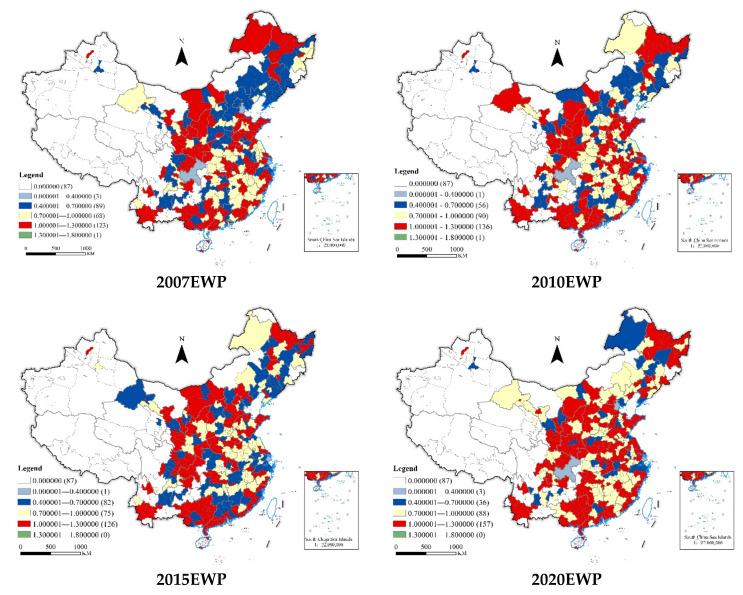
Spatial distribution characteristics of Chinese cities’ EWP from 2007 to 2020. Note: the map is drawn according to the Standard Map Service Website of the Ministry of natural resources (map review No. GS (2020) 4630).

**Figure 6 ijerph-19-12955-f006:**
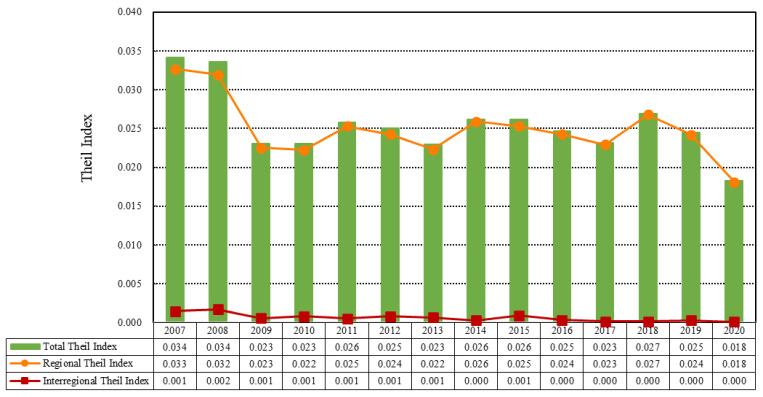
Theil index of cities’ EWP in China from 2007 to 2020.

**Figure 7 ijerph-19-12955-f007:**
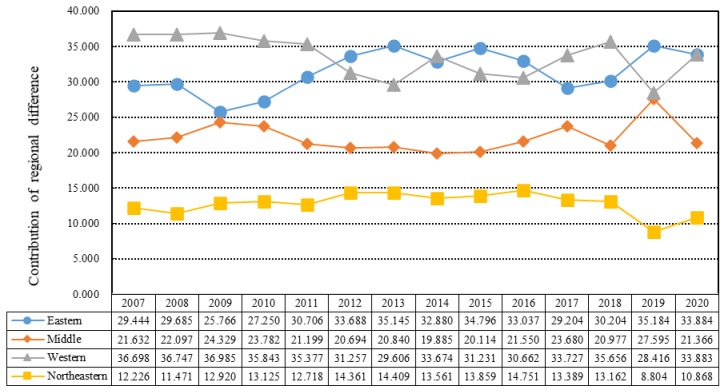
Theil index contribution rate of cities’ EWP in different regions from 2007 to 2020.

**Figure 8 ijerph-19-12955-f008:**
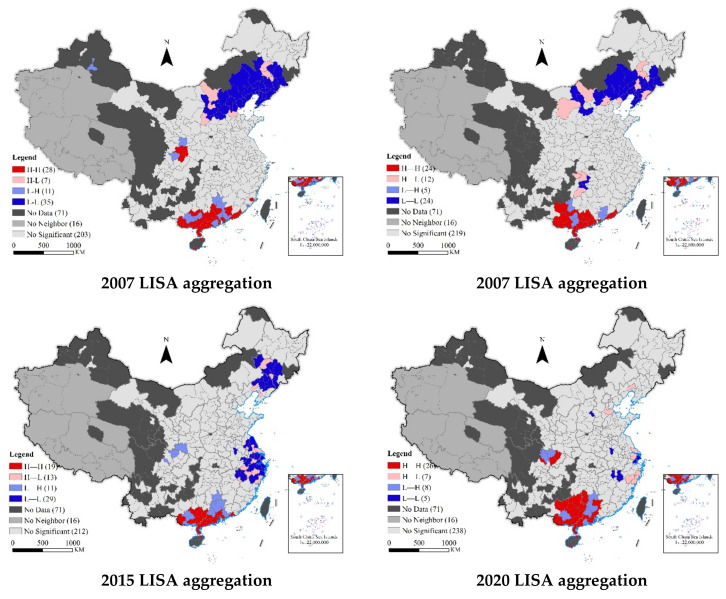
LISA clustering diagram of Chinese cities EWP in 2007, 2010, 2015, and 2020 using the inverse distance matrix. Note: the map is drawn according to the Standard Map Service Website of the Ministry of natural resources (map review No. GS (2020) 4630).

**Table 1 ijerph-19-12955-t001:** Existing EWP studies.

Scale	Authors	Objective Area	Method	Time Period
Provincial Level	Yingjie Feng [15]	30 Chinese provinces	HDI/EF	1994–2014
	Wang et al. [37]	30 Chinese provinces		2006–2018
	Hou et al. [6]	30 Chinese provinces	Super SBM-DEA	2006–2017
	Jing Bian [26]	30 Chinese provinces	Super SBM-DEA	2011–2016
City level	Hu et al. [16]	41 cities	Network DEA	2001–2017
	Liu et al. [32]	171 Prefecture-level cities	Super SBM-DEA	2010–2019
	Xinyi Long [38]	Four islands	HDI/EF	2017
National level	Zhang et al. [33]	82 developed countries		2012
	Sweidan [39]	Gulf countries		1995–2012

**Table 2 ijerph-19-12955-t002:** Indicators of cities’ EWP.

Dimension	First-Level Index Layer	Second-Level Index Layer	Third-Level Index Layer
Input index	Resource input	Energy consumption	Total electricity consumption (100 million kwh)
		Water resource consumption	Water consumption (100 million tons)
		Land resource consumption	Built-up area (square kilometers)
	Non-resource input	Labor input	Number of environmental protection personnel (people)
		Property input	Investment in fixed assets of cities’ public utilities construction (10,000 yuan)
			Environmental protection expenditure (10,000 yuan)
Desirable output	Welfare level	Economic welfare	Cities’ GDP (100 million yuan)
		Environmental welfare	Green space (hectares)
		Social welfare	Years of education (years)
			Number of doctors (people)
			Cities’ road area at the end of the year (10,000 square meters)
Undesirable output	Environmental pollution	Wastewater discharge	Industrial wastewater discharge (10,000 tons)
		Smoke and dust emissions	Industrial smoke and dust (tons)
		Exhaust emissions	Industrial sulfur dioxide emissions (tons)
			Carbon dioxide emissions (tons)

**Table 3 ijerph-19-12955-t003:** Sources of variables.

Variable	Data Sources	Variable	Data Sources
Total electricity consumption	China Urban Statistical Yearbook Statistical yearbook of each city Statistical bulletin and EPS database of each city	Green spaces	China Urban and Rural Construction Statistical Yearbook
Water consumption of the whole society	China Urban and Rural Construction Statistical Yearbook	Three industrial waste products	China Urban Statistical Yearbook
Built-up area		CO_2_	Center for global environmental research
Number of employees in water conservation, environment, and public facilities management	China Urban Statistical Yearbook	Deposits and loans	China Urban Statistical Yearbook
Public assets investment in municipal public facilities construction	China Urban and Rural Construction Statistical Yearbook	Industrial structure	China Urban Statistical Yearbook
Financial expenditure on energy conservation and environmental protection	Financial Bureau and Statistics Bureau apply for information disclosure	General budgetary revenues and expenditures of central and local governments	China Urban Statistical Yearbook China Statistical Yearbook
Per capita years of education	China Urban Statistical Yearbook	Patents	CNRDS database
Number of doctors	China Urban Statistical Yearbook Statistical yearbook of each city	Foreign direct investment	China Urban Statistical Yearbook Statistical yearbooks of provinces and cities
Urban road area at the end of the year	China Urban Construction Statistical Yearbook		

**Table 4 ijerph-19-12955-t004:** Global Moran index of urban EWP in China from 2007 to 2020.

	2007	2008	2009	2010	2011	2012	2013	2014	2015	2016	2017	2018	2019	2020
I	0.042	0.044	0.028	0.033	0.037	0.050	0.031	0.033	0.023	0.008	0.016	−0.003	0.004	0.008
Z	8.798	9.288	6.241	7.043	7.984	10.483	6.757	7.053	5.193	2.173	3.752	0.177	1.513	2.200
P	0.000	0.000	0.000	0.000	0.000	0.000	0.000	0.000	0.000	0.030	0.000	0.859	0.130	0.028

**Table 5 ijerph-19-12955-t005:** Descriptive statistical analysis.

Var Name	Obs	Mean	SD	Min	Median	Max
EWP	3976	0.863	0.192	0.299	0.866	1.798
FINANCE	3976	2.291	1.109	0.560	1.966	8.052
STRU	3976	3.177	10.090	0.107	0.489	67.420
FISCAL REVENUE DEC	3976	0.386	0.165	0.050	0.363	0.834
FISCAL EXPEN DEC	3976	0.789	0.071	0.491	0.799	0.948
LnPATENT	3976	7.202	1.754	1.609	7.100	12.388
FDI	3976	0.018	0.028	0.000	0.011	0.697
URBANIZATION	3976	52.788	15.862	16.413	50.710	100.000

**Table 6 ijerph-19-12955-t006:** LM, LR and Hausman tests of the spatial econometric model.

Test	Statistics	*p*-Value
LM Spatial error	18.389 ***	<0.010
LM Spatial autocorrelation	207.128 ***	<0.010
LR Spatial error	26.67 ***	<0.010
LR Spatial autocorrelation	22.82 ***	<0.010
Hausman	83.61 ***	<0.001

Notes: T-statistics in parentheses, *** denote statistical significance at the 1%, 5%, and 10% level, respectively.

**Table 7 ijerph-19-12955-t007:** Spatial regression results.

Variable	Inverse Distance Matrix		Adjacency Matrix		Economic Distance Matrix	
	Coefficient	t-Value	Coefficient	t-Value	Coefficient	t-Value
FINANCE	0.026 ***	3.469	0.013 *	1.657	0.039 ***	6.279
STRU	0.001 *	1.646	0.001	1.273	0.001	1.348
FISCAL REVENUE DEC	−0.211 ***	−3.029	−0.172 **	−2.423	−0.159 ***	−2.598
FISCAL EXPEN DEC	−0.161	−1.444	−0.114	−0.994	−0.312 ***	−3.175
LnPATENT	0.002	0.271	0.004	0.593	−0.010 *	−1.789
FDI	0.292 *	1.875	0.436 ***	2.640	0.448 ***	3.058
URBANIZATION	0.002 ***	2.812	0.001	1.312	0.002 ***	2.984
FINANCE·W	0.186 ***	2.998	0.055 ***	4.775	−0.001	−0.043
STRU·W	−0.008	−1.374	−0.001	−0.525	−0.002	−1.083
FISCAL REVENUE DEC·W	1.363 **	2.353	0.102	0.979	−0.256	−1.531
FISCAL EXPEN DEC·W	−0.446	−0.511	−0.249	−1.445	0.517 **	2.101
LnPATENT·W	−0.067	−1.346	−0.030 ***	−3.151	0.004	0.257
FDI·W	2.150	1.418	0.181	0.648	2.401 ***	3.930
URBANIZATION·W	−0.006	−1.420	0.003 ***	3.041	−0.001	−0.329
Time fixed	Yes		Yes		Yes	
Individual fixed	Yes		Yes		Yes	
ρ	0.343 ***	2.862	0.093 ***	4.212	−0.058 *	1.793
sigma2_e	0.014 ***	44.560	0.014 ***	44.546	0.014 ***	44.575
Observations	3976 3976		3976 3976		3976 3976	

Note: *, **, and *** are significant at 10%, 5% and 1% respectively.

## Data Availability

Data available on request.

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
