# Peer review of "The Spatiotemporal Evolution and Influencing Factors of the Chinese Cities’ Ecological Welfare Performance"

_ijerph, 2022, doi:10.3390/ijerph191912955_

Round 1
Reviewer 1 Report
The paper is generally written well and is understandable although I think the scope is relatively too broad. Nonetheless, the work presented is meaningful and I have a few comments to help improve the standard of the present manuscript. My comments are summarised below.
#Last Line in First Page: Page 1 of 24: 'The improvement of EWP is the core to solve the problems of economic growth, ecological protection and improvement of people's livelihood'.
COMMENT: Even though the above sentence is written well, the abbreviation EWP has to be written in full first before you can use it like in that sentence. I see that you have included the full meaning of EWP term in your Abstract. However, can you please do this again in the respective sentence.
COMMENT: The usage of a semi colon ( : ) between several sentences that appear to be independent is unacceptable. Can you please check the following sentence below specifically to correct the inappropriate use of a semi colon. Unfortunately, this problem occur in many parts of this manuscript. Please do the adjustments.
'At present, China's EWP level is still low, at the level of economic development, according to the open data of the International Monetary Fund (IMF), China's per capita GDP in 2021 is US $12359, ranking 63 in the world, which is still far from developed countries; At the level of environmental protection, a large number of pollutants generated by economic activities have become the bottleneck restricting China's economic development. China's carbon emissions still account for a high proportion, and sulfur dioxide emissions, water pollution are not optimistic [7]; At the level of welfare promotion, a series of social problems such as poverty, unemployment, education and health have emerged in the process of urbanization, which have restricted China's economic development [8}.
COMMENT: The quality of Figure 1 is not good in terms of resolution. The information is not readable especially for the legend that accompany the map. Plese use a better resolution so that we can read with ease thereby enabling us to interpret the map provided.
COMMENT: The results are first presented with various graphical representations which is fine but then the Discussion of results happens in the Conclusion part. My suggestion is that the Discussion of Results should be relocated and placed immediately after the presentation of results.
COMMENT: Earlier in the introduction section, the authors wrote as follows:
The purposes of this paper are: (1) to establish an index system of EWP and measure Chinese urban EWP; (2) To explore the temporal and spatial evolution of Chinese urban EWP and the main sources of EWP differences; (3) Explore the spatial correlation and driving factors of urban EWP in China
However, when it comes to the writing of the conclusion, it is not clear how all of the three research objectives are accounted for. My suggestion is that you must use words like Firstly, Secondly, and then Lastly so that proper explanation is made for each respective conclusion point.
COMMENT: In the section where conclusions are being provided, there are no discussions on what are the implications of the findings for policy development in the study area. In other words, there is a need to explain how policy development can benefit from the results.
COMMENT: In the present section that deal with Conclusion, there are no further recommendations regarding what future research is being suggested. Can you please provide further recommendations to guide future research directions.
Hoping the comments are understandable and can improve the quality of the manuscript.
Author Response
The Spatiotemporal Evolution and Influencing Factors of Chinese Urban Ecological Welfare Performance
Reviewer1
The paper is generally written well and is understandable although I think the scope is relatively too broad. Nonetheless, the work presented is meaningful and I have a few comments to help improve the standard of the present manuscript. My comments are summarised below.
COMMENT: Last Line in First Page: Page 1 of 24: 'The improvement of EWP is the core to solve the problems of economic growth, ecological protection and improvement of people's livelihood'. Even though the above sentence is written well, the abbreviation EWP has to be written in full first before you can use it like in that sentence. I see that you have included the full meaning of EWP term in your Abstract. However, can you please do this again in the respective sentence.
Answer: Thank you for your careful review. According to your suggestion, we have marked the full name of the ecological welfare performance in the introduction for the first time and pointed out that the abbreviation EWP is used later. See line 48 of the introduction for details.
- COMMENT: The usage of a semi colon ( : ) between several sentences that appear to be independent is unacceptable. Can you please check the following sentence below specifically to correct the inappropriate use of a semi colon. Unfortunately, this problem occur in many parts of this manuscript. Please do the adjustments. 'At present, China's EWP level is still low, at the level of economic development, according to the open data of the International Monetary Fund (IMF), China's per capita GDP in 2021 is US $12359, ranking 63 in the world, which is still far from developed countries; At the level of environmental protection, a large number of pollutants generated by economic activities have become the bottleneck restricting China's economic development. China's carbon emissions still account for a high proportion, and sulfur dioxide emissions, water pollution are not optimistic[7]; At the level of welfare promotion, a series of social problems such as poverty, unemployment, education and health have emerged in the process of urbanization, which have restricted China's economic development [8}.
Answer: Thanks to the reviewer for their careful review. We have checked the whole article for improper semicolons and made modifications according to your suggestions. see line 50-60, line 61-70.
- COMMENT: The quality of Figure 1 is not good in terms of resolution. The information is not readable especially for the legend that accompany the map. Please use a better resolution so that we can read with ease thereby enabling us to interpret the map provided.
Answer: Thank the reviewers for their careful review. We have modified Figure 1 to improve its resolution, ensure its readability, and checked and revised the rest of the text. See Figure 1—Figure 8 for details.
- COMMENT:The results are first presented with various graphical representations which is fine but then the Discussion of results happens in the Conclusion part. My suggestion is that the Discussion of Results should be relocated and placed immediately after the presentation of results.
Answer: Thank the reviewers for their careful review. We have replaced the discussion and conclusion according to your suggestions, and adjusted the discussion part to the result. For details, please refer to the fifth part, Discussion and Conclusion.
- COMMENT: Earlier in the introduction section, the authors wrote as follows:
The purposes of this paper are: (1) to establish an index system of EWP and measure Chinese urban EWP; (2) To explore the temporal and spatial evolution of Chinese urban EWP and the main sources of EWP differences; (3) Explore the spatial correlation and driving factors of urban EWP in China
However, when it comes to the writing of the conclusion, it is not clear how all of the three research objectives are accounted for. My suggestion is that you must use words like Firstly, Secondly, and then Lastly so that proper explanation is made for each respective conclusion point.
Answer: Thank the review experts for their careful review. According to your suggestion, we use the first and second ordinal words in the conclusion part to present the conclusion of this paper more clearly. Specifically, we first outline the research framework of this paper, and then present the spatio-temporal evolution characteristics, regional differences and sources, spatial correlation and influencing factors of China's cities’s EWP in turn according to the research purpose. See Section 5.2 Conclusion for details.
- COMMENT: In the section where conclusions are being provided, there are no discussions on what are the implications of the findings for policy development in the study area. In other words, there is a need to explain how policy development can benefit from the results.
Answer: Thank the review experts for their careful review. Based on your suggestions, we propose corresponding policy suggestions based on the results. First of all, according to conclusion 1, China's EWP is weakly ineffective, and the spatial distribution is uneven. We propose the following policy recommendations:
In the future, it is necessary to further improve the EWP to an effective level. First, we need to transform the GDP oriented development model, promote the conversion of new and old driving forces, and improve resource utilization by implementing the innovation driven development strategy.Second, we should pay attention to the improvement of social welfare, promote the equalization of public services, and improve people's living standards. By improving the management level and technology, the scale technical efficiency, pure technical efficiency and the EWP will be improved.
At the level of narrowing the regional differences in EWP, all regions should base on the development stage of the region and coordinate to improve the EWP according to local conditions. The spatial correlation of EWP should be called an important reference for the government to formulate development policies. Actively cooperate with other cities to break regional administrative barriers. While improving their own EWP, central cities will drive the development of other cities with low EWP. Cities with lowEWP should consciously benefit from the spillover effects of other cities to improve urban EWP. The eastern region has the highest pure technical efficiency. It has high-quality human capital and scientific research institutes that absorb international technology spillovers. The eastern region should focus on promoting social equity and coordinated social development. The central region should pay attention to industrial upgrading and transformation, introduce advanced technology, and make up for the shortcomings of people's livelihood. The western region should further expand the level of opening up on the basis of maintaining ecological advantages and actively integrate into the the Belt and Road initiative. Northeast China, which has the lowest EWP, should further improve the opening-up coordination mechanism and create a characteristic industrial chain to improve the technology and management level.
According to conclusion 3, there is a significant positive spatial correlation between China's EWP, and financial development level, industrial structure, fiscal revenue decentralization, opening up level, and urbanization level have a significant impact on the EWP. Based on this, in the future, China should use the spillover effect of different influencing factors to promote the improvement of EWP.
Give play to the scale effect of finance and promote technology spillovers. As financial development plays an obvious role in promoting EWP, all regions should speed up the reform of financial system, constantly stimulate the innovation of financial products, leverage social capital into green production, and play the role of finance in promoting EWP. The concept of green development should be applied throughout the whole process of industrial restructuring to achieve sustainable development of the ecological environment and high-quality coordination of the economy. We will promote environmental protection and technological upgrading in the secondary industry and increase productivity. China should vigorously develop new energy, new materials, high-end equipment manufacturing and other emerging industries, promote the green transformation of the manufacturing industry, and create a green and ecological manufacturing industry cluster. Promote the upgrading of the digital enabling industrial structure, and promote the low-carbon transformation of the industry with digital technology. China should expand the level of opening up to the outside world, and adhere to the pattern of opening up from east to west and by land and sea. According to cities’s own economic development level, social and public needs, and regional factors to select the industry attributes, investment scale, and equipment of foreign investment, implement industrial policy support and tax incentives for cleaner production enterprises, encourage the promotion of research and development technology, and give play to the scale and spillover effects of foreign investment. In the process of urbanization, local governments should adhere to the concept of green development, give consideration to the coordination of land expansion, infrastructure and greening construction, and pay attention to the coupling and coordination of economic benefits and ecological efficiency. Accelerate the new urbanization process with county as the carrier, form a new urbanization pattern with regional linkage, reasonable structure, intensive and efficient, green and low-carbon, and promote the improvement of EWP.
- COMMENT: In the present section that deal with Conclusion, there are no further recommendations regarding what future research is being suggested. Can you please provide further recommendations to guide future research directions.
Answer: Thank the review experts for their careful review. In the last paragraph of 5.1 discussion, we pointed out the shortcomings of this paper and the future research direction. Specifically:
The limitation of this study is that in the construction of the index system, this paper adopts the objective EWP evaluation method, ignoring the subjective feeling and cognitive evaluation of individual perception. In the future, the subjective and objective evaluation can be integrated into an index system to comprehensively evaluate the urban EWP. This paper only makes spatial econometric analysis on the influencing factors of EWP, and does not further explore the path that affects EWP. In the future, we can select a certain aspect at the government or the social level to further explore the specific path that affects EWP. We can also use the causal inference method to explore the net impact effect of a policy on EWP, and enrich the research on EWP.
According to the suggestions of the reviewers, we embellished the full text of the language expression and marked it in the text in a revised mode.
Thanks again for the hard work of the editing teachers and reviewers!
Thank you again for your careful review and relevant suggestions.

Reviewer 2 Report
In this paper, the spatiotemporal evolution and influencing factors of the ecological welfare performance of 284 cities in China were measured. In which the improved evaluation index system for ecological welfare performance was established and multiple analysis methods were used to assess its characteristics, which has some practical implications. However, from the full text, there are still some problems and concerns:
(1) My main concern is the contribution of this research to this field, which is less prominent than previous studies. The biggest innovation of this paper is to improve the input-output index system of ecological welfare performance. Nevertheless, different scholars consider the concept of ecological welfare performance from different perspectives. How can the author assure that his index system is scientific and credible? It is recommended to provide literature for further demonstration.
(2) In a literal sense, ecological welfare performance reflects more ecological meaning. Why is the concept of ecological welfare performance better than other indexes? The index that reflects the coupling of resources, economy and ecology may better reflect the concept of sustainability development. In Section 2, The Human Development Index can be seen as only a partial picture of what sustainable development means rather than ecological welfare performance.
(3) Various analysis methods have been applied to study the spatial and temporal evolution characteristics of ecological welfare performance, which undoubtedly improves the richness and level of the research content. However, the authors should pay more attention to whether the methods are well applicable to the research content of this paper and whether they can explain the phenomena existing in reality or reveal the general rule, instead of piling up methods blindly. It is recommended to add more references to prove its rationality.
(4) The quality of the figures needs to be further refined. ​First, they need to be more normative, and second, they need to be clearer. It is difficult to judge the author’s research results based on the existing figures.
(5) There are some criteria for classifying research results (e.g., Pages 10 and 12). The standard division is very subjective, different criteria would lead to different research results. It is necessary to give the basis of the standard division.
(6) Results and discussion are presented with figures and more or less a description, while better interpretation and practical implications should be added in the above two aspects. In particular, phrases like “possibly because” should be eliminated.
(7) Regarding the research content and data collection of the article, “urban” in the title should be altered to “city”.
Author Response
The Spatiotemporal Evolution and Influencing Factors of Chinese Urban Ecological Welfare Performance
Reviewer 2
- COMMENT: In this paper, the spatiotemporal evolution and influencing factors of the EWP of 284 cities in China were measured. In which the improved evaluation index system for EWP was established and multiple analysis methods were used to assess its characteristics, which has some practical implications. However, from the full text, there are still some problems and concerns:
- My main concern is the contribution of this research to this field, which is less prominent than previous studies. The biggest innovation of this paper is to improve the input-output index system of EWP. Nevertheless, different scholars consider the concept of EWP from different perspectives. How can the author assure that his index system is scientific and credible? It is recommended to provide literature for further demonstration.
Answer: Thank the review experts for their careful review. According to your suggestion, we have supplemented relevant literature in 3.1 Index Construction from the perspective of input and output index selection. The details are as follows:
In terms of selection of input indicators, natural capital in the ecosystem directly or in-directly contributes to human welfare[43],Most of the existing studies have included natural resource input, such as water resource consumption, land resource utilization and power consumption[44]. In addition to the above indicators, this paper also includes labor and asset input into the indicator system. The reason is that EWP is the efficiency of transforming natural consumption into welfare level, in which the ecosystem can provide direct ecological services, but it can only be transformed into material wealth through a series of links such as production, processing, consumption and distribution to meet the needs of human welfare [16].Xiao et al.also believe that ecological input should not only cover resource indicators, but also introduce capital indicators to comprehensively reflect the regional EWP[45].This paper selected energy consumption, water resource con-sumption, land resource consumption, labour input and property input, which were measured by the social electricity consumption, water consumption, built-up area, the number of environmental protection personnel, the investment in fixed assets of urban public utilities and the environmental protection expenditure. It is worth noting that in the input index, this paper not only selected the input of ecological resources, but also in-cluded the input of labour and property[16], and this approach is different from most existing studies. Urban public utilities are important parts of the urban infrastructure, which play an important role in ensuring the normal operation of the city and improving the living environment. The main investments include water supply, gas, rail transit, drainage and landscaping, which will affect the scale, speed and healthy development of the city[46], therefore, they were included in the index.
On the output level, this paper divides it into desirable output and undesirable output. According to Hu et al[16], dividing welfare into three categories: economic welfare, social welfare and environmental welfare[47][48].In the undesirable output, this paper not only includes the industrial three wastes into the evaluation system, but also includes the carbon dioxide emissions. Carbon dioxide is usually called a greenhouse gas.and its increase in emissions is the main reason for global temperature rise. At the same time, carbon dioxide emissions are also considered one of the reasons for health risks. Betti et al. regard carbon dioxide emissions as an objective environmental variable that affects na-tional welfare [49], Yasin et al. also measure environmental well-being based on carbon dioxide emissions [50].
In the selection of output indicators, this paper selected welfare level as the desirable output and environmental pollution as the undesirable output. Among them, welfare output includes economic welfare, social welfare and environmental welfare, which were measured by urban GDP, park green area, average education years, number of doctors and actual road area. Environmental pollution was measured by three kinds of industrial waste discharge and carbon dioxide emissions. Regarding the global efforts to achieve carbon neutrality, it is necessary to include carbon dioxide emissions in the indicator system. It should be noted that adding variables to the DEA model will result in higher weight space dimensions and efficiency scores[51]. That is, when the number of DMUs is fixed, the more input—output indicators included in the model, the greater the efficiency value obtained, and the lower the identification of DEA analysis[52]. Therefore, the variable selection of DEA model should be as few as possible, and the practice of existing research should be used for reference[53]. In this paper, the entropy method is used to take the combined pollution index of industrial three wastes and carbon dioxide emissions as the undesirable output.
- COMMENT: In a literal sense, EWP reflects more ecological meaning. Why is the concept of ecological welfare performance better than other indexes? The index that reflects the coupling of resources, economy and ecology may better reflect the concept of sustainability development. In Section 2, The Human Development Index can be seen as only a partial picture of what sustainable development means rather than ecological welfare performance.
Answer: Thank the review experts for their careful review. Why is the concept of ecological welfare performance better than other indicators proposed by reviewers? The index reflecting the coupling of resources, economy and ecology may better reflect the concept of sustainable development. We answered as follows:
â‘ The EWP mentioned in this paper is the coupling of economy, ecology and welfare, while the previous indicators such as ecological efficiency and economic performance only emphasize a certain level, belonging to the category of weak sustainable development, while the EWP covers a wider range, and the measurement results can reflect the level of sustainable development more. See 2 Literature Review for specific modifications:
Ecological efficiency and economic performance are all concepts of sustainable development. Although they are similar to EWP, these concepts are still research paradigms of weak sustainable development.Only focusing on the economic growth brought by unit resource input cannot comprehensively measure sustainable development [13] [14]. EWP is more abundant than ecological efficiency because it takes into the account natural environment constraints and social welfare on the basis of traditional economic efficiency guidance[15].EWP is an upgraded version of ecological efficiency. Based on the evaluation of ecological efficiency, EWP is integrated into human development, connecting the three major systems of economy, society and ecology[16], It emphasizes the development concept of decoupling ecological resource consumption from social welfare, breaking through the limitations of traditional GDP in measuring the quality of human life[17].EWP can reflect local governance level and people's happiness[18] and it can provide a new perspective and analysis tool for strong and sustainable urban development[19].EWP is a people-oriented concept of sustainable development, which helps to achieve cleaner production and improve the coupling and coordination of economic system, ecological system and social system[20].
The human development index proposed by the reviewers can only be regarded as a part of what sustainable development means, rather than the performance of ecological welfare. We answered as follows:
â‘¡According to the opinions of reviewers, we re sorted out the human development index to measure the ecological welfare performance, and pointed out the shortcomings of using the ratio of human development index and ecological footprint to measure the EWP. See 2 Literature review for details:
As a component of the HDI, economic development has improved the technology and ability to deal with natural disasters and the level of human education, which will cover up the consequences of ecological deterioration.Therefore, measuring human welfare with HDI will enlarge the level of human welfare [24]. The ratio method is suitable for analyzing independent and discontinuous objects, especially single projects and technologies, while the indicator system is suitable for analyzing the coordinated development of multiple systems[25]
- COMMENT: Various analysis methods have been applied to study the spatial and temporal evolution characteristics of ecological welfare performance, which undoubtedly improves the richness and level of the research content. However, the authors should pay more attention to whether the methods are well applicable to the research content of this paper and whether they can explain the phenomena existing in reality or reveal the general rule, instead of piling up methods blindly. It is recommended to add more references to prove its rationality.
Answer: Thank the review experts for their careful review. According to your suggestion, we have supplemented the logic and reason for method selection. See 3.2 Research Methods for details. Specific contents are as follows:
Research methods are based on the applicability of research questions. This paper first explores the spatio-temporal evolution trend of EWP, so we first use Super SBM-DEA method to measure urban EWP.DEA method can solve the problem of inconsistent input of various resources and output units of environmental pollution, and it does not need to consider specific production functions, weights and parameters. It breaks through the limitation that the upper limit of the traditional efficiency. When the DMUs reach the efficiency boundary, the efficiency values of the DEA can be reordered. After calculation, it is found that there is a large difference in EWP among different cities, so we use the Theil index to observe the source of the difference in EWP. With the rapid economic growth after China's reform and opening up, regional differences have become a major challenge in China. The regional balance of EWP is of great significance to China's sustainable development. The commonly used indicators for assessing regional differences or inequality can be divided into three categories: coefficient of variation, Lorentz curve index, entropy or information theory index. The coefficient of variation is easy to calculate, but sensitive to outliers. The Gini coefficient is an index based on the Lorentz curve, but it is easily affected by high values. One of the advantages of the Theil index is that it can be decomposed into additive terms to describe the inequality between and within elements in the system[54].The advantage of Theil index is that the overall regional differences can be decomposed into inter—regional differences and intra—regional differences, which can better reflect the regional imbalance in the heterogeneous regional structure[55].In order to further improve the level of EWP, it is necessary to explore the factors that affect the EWP. The impact of spatial effect cannot be ignored. Therefore, this paper first uses spatial exploratory analysis to test the spatial correlation of EWP. Exploratory Spatial Data Analysis (ESDA) is a spatial data analysis method, which can intuitively and clearly represent the spatial correlation and aggregation of geographical elements, and is widely used in the spatial analysis of elements [56].Generally, the global spatial autocorrelation is used to represent the spatial correlation of geographical phenomena in the whole region, and the local spatial autocorrelation is used to reflect the spatial concentration of certain elements. The spatial measurement method is used to explore the factors that affect the EWP and provide empirical evidence for improving the EWP.
- COMMENT: The quality of the figures needs to be further refined. ​First, they need to be more normative, and second, they need to be clearer. It is difficult to judge the author’s research results based on the existing figures.
Answer: Thank the review experts for their careful review. According to your suggestions, we have supplemented and sorted out the data sources of the paper. See 3.3 Data Sources and Regional Division for details.
data sources of measuring EWP and influencing factors is shown below table. In terms of data sources, the following points need to be explained: First,What needs to be explained is the data collection of smoke and dust emissions and carbon dioxide emissions: in 2020, the statistical reporting system was significantly adjusted, and compared with the 13th five-year plan statistical reporting system, the statistical caliber and other aspects were changed. Smoke and dust were renamed particulate matter. Second,at the level of carbon dioxide emission data, some scholars try to use night light data to retrieve the carbon emission footprint of municipal or lower administrative regions. However, due to the defects of the light data itself, such as background noise and discontinuity, its application scenarios have not been widely accepted. Carbon dioxide emission data are from the open anthropogenic carbon dioxide (ODIAC) fossil fuel emission data set of the Global Environmental Research Center(https://db.cger.nies.go.jp/dataset/ODIAC/).ODIAC first introduced the combination of night lighting data and emission/location profile of a single power plant to estimate the spatial range of carbon dioxide emissions from fossil fuels. The spatial resolution is 1-KM, and the unit is t/KM2. The product is generated by combining multi-source night lighting data, global point source database and ship/aircraft fleet track. The data can match the global, regional and urban CO2 emissions and meet the requirements of large-scale and long-term series[62].The open-source data inventory for anthropogenic CO2 (ODIAC) data is the monthly data set of carbon emission spatial grid derived by ODA and other countries. The annual statistical data of countries are used for spatial and temporal decomposition to obtain the global monthly spatial grid data [63]. By cutting, synthesizing and extracting China’s carbon emission grid data, extracting Chinese carbon emission grid data and synthesizing the annual data set, the carbon dioxide emissions of China’s prefecture level cities from 2007 to 2019 are sorted out, of which the carbon dioxide emissions in 2020 are supplemented by trend function. In terms of data processing, This paper synthesizes four environmental pollutants into an environmental pollution index. For missing data in individual years, using trend function and interpolation method to complete. in order to eliminate the influence of price factors, the data related to prices in the full text are treated with constant prices based on 1978 prices.
- COMMENT: There are some criteria for classifying research results (e.g., Pages 10 and 12). The standard division is very subjective, different criteria would lead to different research results. It is necessary to give the basis of the standard division.
Answer: Thank the review experts for their careful review. The reasons for our classification of EWP standards are as follows:
Considering that the natural breakpoint method in ArcGIS cannot directly compare the EWP in different years, this paper uses the natural breakpoint method and the equal division method of ArcGIS to grade the EWP. If the EWP exceeds 1.0, it means it is effective; if it is less than 1.0, it is invalid. In the analysis of the effective range, this paper also takes 1.3 as the dividing point, and those exceeding 1.3 are strong effective, while those not exceeding 1.3 are weak effective. In the analysis of the invalid range, 0.4 and 0.7 are the dividing points, less than 0.4 is strong invalid, 0.4-0.7 is invalid, and 0.7-1.0 is weak invalid.
- COMMENT: Results and discussion are presented with figures and more or less a description, while better interpretation and practical implications should be added in the above two aspects. In particular, phrases like “possibly because” should be eliminated.
Answer: Thank the review experts for their careful review. In the discussion section, we have added dialogues with existing research. See the text for details. According to your suggestion, we revised the full text and deleted the phrase "possibly because".
- COMMENT: Regarding the research content and data collection of the article, “urban” in the title should be altered to “city”.
Answer: Thank the review experts for their careful review. According to your suggestion, we replaced “urban” with “city” in the full text.
According to the suggestions of the reviewers, we embellished the full text of the language expression and marked it in the text in a revised mode.
Thanks again for the hard work of the editing teachers and reviewers!
Thank you again for your careful review and relevant suggestions.

Round 2
Reviewer 2 Report
Thanks for the revisions, which addressed all my comments. The MS has been substantially improved by properly handling the comments. My final concern is whether the word “Cities’s” in the title is used correctly, please confirm further.